# Text2Arch: A Dataset for Generating Scientific Architecture Diagrams from Natural Language Descriptions

**Shivank Garg**
IIT Roorkee, India
`shivank_g@mfs.iitr.ac.in`

**Sankalp Mittal**
Google, India
`sankalpmittal123@gmail.com`

**Manish Gupta**
Microsoft, India
`gmanish@microsoft.com`

## Abstract

Communicating complex system designs or scientific processes through text alone is inefficient and prone to ambiguity. A system that automatically generates scientific architecture diagrams from text with high semantic fidelity can be useful in multiple applications like enterprise architecture visualization, AI-driven software design, and educational content creation. Hence, in this paper, we focus on leveraging language models to perform semantic understanding of the input text description to generate intermediate code that can be processed to generate high-fidelity architecture diagrams. Unfortunately, no clean large-scale open-access dataset exists, implying lack of any effective open models for this task. Hence, we contribute a comprehensive dataset, Text2Arch, comprising scientific architecture images, their corresponding textual descriptions, and associated DOT code representations. Leveraging this resource, we fine-tune a suite of small language models, and also perform in-context learning using GPT-4o. Through extensive experimentation, we show that Text2Arch models significantly outperform existing baseline models like DiagramAgent and perform at par with in-context learning based generations from GPT-4o. We make the code, data and models publicly available[1].

## 1 Introduction

In an era where complex systems are increasingly described, designed, and communicated through natural language, the ability to automatically translate textual descriptions into precise, semantically faithful architecture diagrams holds transformative potential. Manual diagram creation is time-consuming and can be error-prone. A system which can convert input scientific text descriptions to architecture images could be useful for many applications as follows. Authors could use it translate their method descriptions to architecture block diagrams automatically enabling high quality software documentation, academic research documents, and patent drafts. A text to architecture system can bridge the gap between textual design intent and visual representation, enabling end-to-end AI-assisted software engineering. Usage of such systems for enterprise architecture visualization can help in quick and effective understanding of complex enterprise systems. Broadly, such systems can enable automatic generation of educational diagrams from lesson text, enhancing visual learning. Automatic diagram creation tools could also help in quick updates to existing diagrams as text descriptions evolve. Overall text-to-architecture systems have the potential to significantly boost productivity in fields such as education, scientific research, and industry, where clear and structured visual representations are crucial for effective communication and analytical reasoning. They can accelerate ideation, improve collaboration, and reduce ambiguity in technical communication.

---

[1]Code:      https://github.com/shivank21/text2arch;      Models:      https://huggingface.co/shivank21/text2arch-deepseek/; Data: https://huggingface.co/datasets/shivank21/text2archdata

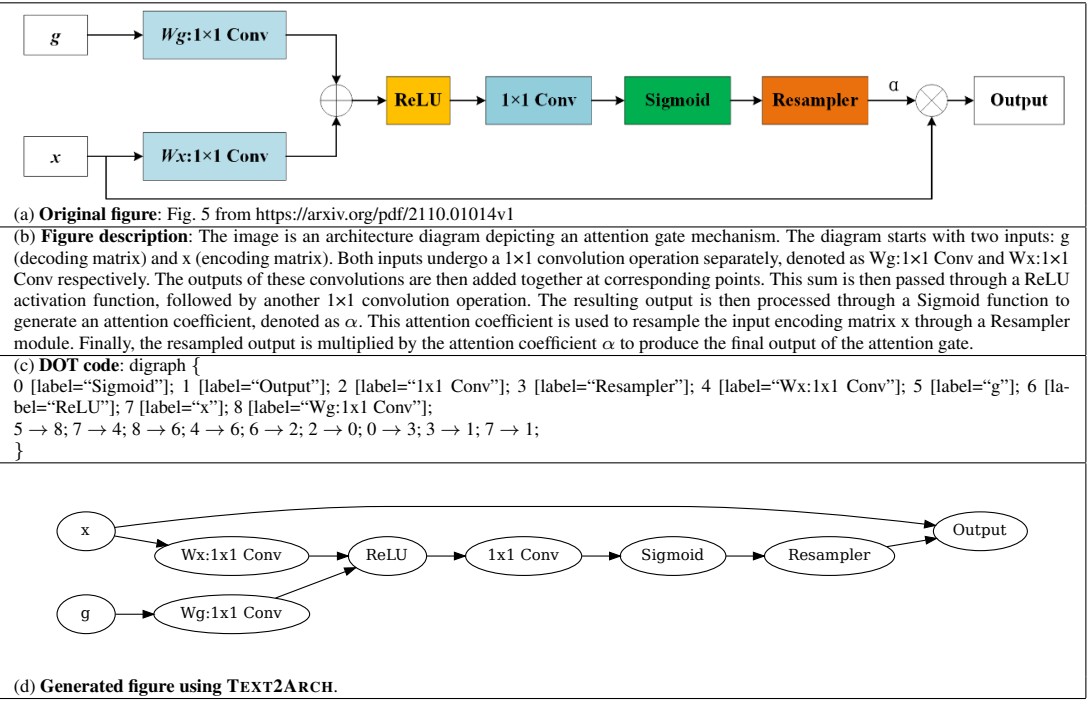

(a) **Original figure**: Fig. 5 from https://arxiv.org/pdf/2110.01014v1

(b) **Figure description**: The image is an architecture diagram depicting an attention gate mechanism. The diagram starts with two inputs: g (decoding matrix) and x (encoding matrix). Both inputs undergo a 1×1 convolution operation separately, denoted as Wg:1×1 Conv and Wx:1×1 Conv respectively. The outputs of these convolutions are then added together at corresponding points. This sum is then passed through a ReLU activation function, followed by another 1×1 convolution operation. The resulting output is then processed through a Sigmoid function to generate an attention coefficient, denoted as $\alpha$. This attention coefficient is used to resample the input encoding matrix x through a Resampler module. Finally, the resampled output is multiplied by the attention coefficient $\alpha$ to produce the final output of the attention gate.

(c) **DOT code**: digraph {
0 [label="Sigmoid"]; 1 [label="Output"]; 2 [label="1x1 Conv"]; 3 [label="Resampler"]; 4 [label="Wx:1x1 Conv"]; 5 [label="g"]; 6 [label="ReLU"]; 7 [label="x"]; 8 [label="Wg:1x1 Conv"];
5 → 8; 7 → 4; 8 → 6; 4 → 6; 6 → 2; 2 → 0; 0 → 3; 3 → 1; 7 → 1;
}

(d) **Generated figure using TEXT2ARCH**.

Figure 1: An example from TEXT2ARCH dataset. (a) shows the original figure while (d) shows the automatically generated figure using our proposed TEXT2ARCH model. TEXT2ARCH takes the figure description (b) to generate the DOT code (c) which is then compiled to obtain (d). Fig. (d) can be easily modified by a human expert to achieve the correct diagram.

Yet, despite the growing power of large language models (LLMs), the task of generating high-fidelity scientific architecture diagrams from text is challenging. Unlike natural scene images, diagram generation demands strict semantic alignment, structural coherence, and fine-grained precision. The task remains largely unexplored, primarily due to the absence of high-quality datasets and robust modeling baselines. This paper addresses that gap. In this work, we focus on the novel problem of *text-to-architecture diagram generation (*TEXT2ARCH*)*, where the goal is to generate architecture diagrams, composed of labeled nodes and directed edges, directly from natural language figure descriptions.

In recent years, text-to-image models (Nichol et al., 2021; Vahdat & Kautz, 2020; Reed et al., 2016), particularly diffusion models (Song et al., 2020; Rombach et al., 2022; Saharia et al., 2022), have significantly advanced image generation, enabling the generation of highly realistic images for various industrial applications from simple text prompts (Capogrosso et al., 2024; Li et al., 2024). However, they are inherently limited when it comes to structured architecture diagram generation (Rodriguez et al., 2023b;a; Zala et al., 2023). These models often suffer from short input context windows, e.g., Stable Diffusion's use of CLIP restricts input to 77 tokens, and cannot adequately process or reason over long textual inputs. Approaches like LongAlign (Liu et al., 2024b) aim to extend input capabilities, but the core architectural limitations remain. Moreover, diffusion-based models struggle to capture explicit logical structures, often producing diagrams with unreadable or incorrect textual components and poorly organized visual elements. Lastly, it is difficult to edit and further refine such generated images.

Hence, alternative techniques have attempted to convert text descriptions to intermediate code in a graph description language (e.g., TikZ), and then render the code into diagrams using standard compilers like DOT[2] and TikZ[3]. While effective for simple plots and charts, these methods falter in generating semantically rich and hierarchically organized diagrams. Recent efforts such as Diagra-

---

[2]https://www.graphviz.org/documentation/
[3]https://github.com/pgf-TikZ/pgf/

mAgent (Wei et al., 2024) improve diagram synthesis, editing, and reasoning by incorporating both textual and visual modalities. We follow this line of work. However, unlike DiagramAgent, which uses a multi-agent framework and a broad benchmark spanning eight diagram types with loosely aligned text-code-image pairs, our proposed TEXT2ARCH system focuses exclusively on scientific architecture diagrams with clean, semantically aligned triplets of textual descriptions, DOT code, and images. Additionally, TEXT2ARCH adopts a streamlined end-to-end approach and introduces rich set of novel graph-level evaluation metrics, offering deeper insights into structural fidelity and making it more practical for real-world use.

A core bottleneck in this field is the lack of a large-scale, high-quality, open-access dataset that pairs detailed textual descriptions with corresponding architecture diagrams and their structured code representations. Existing datasets like ACL-Fig (Karishma et al., 2023) and Paper2Fig (Rodriguez et al., 2023b) include a mix of diagram types, often lack consistent labeling, and do not contain clean, well-aligned textual descriptions specific to architectural content. To address this gap, we introduce TEXT2ARCH, a large-scale benchmark dataset consisting of $75,127$ samples of architecture diagram images, their corresponding clean textual descriptions and the corresponding DOT code representations. We leverage GPT-4o prompting, structured diagram parsers as well as relevant paragraphs from paper pdfs to obtain clean textual descriptions and the DOT code representations. The dataset is divided into $60,519$ train, $7565$ validation and $7043$ test samples.

We perform zero-shot inference using GPT-4o as well as smaller models like Qwen2-7B (Qwen Team, 2024), DeepSeek-llm-7b-base (Liu et al., 2024a) and Meta-Llama-3-8B (Dubey et al., 2024). Using the train set of TEXT2ARCH, we develop a family of LLM-based models (in the 7B-8B parameter range) fine-tuned specifically for the task of generating structured DOT code from natural language input. We also compare the performance with DiagramAgent (Wei et al., 2024). We report performance on the test set of TEXT2ARCH as well as on a human labeled measurement set of 99 images.

Although our end goal is to generate architecture diagrams, TEXT2ARCH generates the DOT code post which we use the standard DOT compiler to generate the image. Hence, it is not very meaningful to evaluate the quality of the image. Instead we measure the output quality using two sets of metrics: standard natural language generation (NLG) metrics and graph based metrics. As part of the standard NLG metrics, we use sequence similarity (ROUGE-L), structural and semantic alignment (CodeBLEU), and edit-based distance (Levenshtein) between the generated DOT code and the ground truth code. We also include character-level overlap (chrF) to capture nuanced differences between predicted and reference code. Graph based metrics are based on graph representations of predicted and ground-truth diagrams. They include similarity-weighted node precision, recall, and F1-score using optimal node matching via the Hungarian algorithm (Kuhn, 1955). We also extend the evaluation to edges through precision, recall, and Jaccard similarity. Additionally, we propose Node PR-AUC by varying the matching threshold to assess robustness across similarity levels.

We make the following main contributions in this paper. (1) We introduce the novel task of generating scientific architecture diagrams from natural language descriptions via intermediate DOT code. (2) To support this, we release TEXT2ARCH, a large-scale dataset of over 75K samples containing architecture images, clean textual descriptions, and corresponding DOT code. Samples are also divided into easy, medium and hard buckets. (3) We fine-tune multiple small language models and evaluate GPT-4o for this task, showing significant improvements over existing baselines like DiagramAgent. (4) Our evaluation framework includes both standard NLG and graph-level metrics to rigorously assess semantic and structural fidelity.

## 2 RELATED WORKS

### 2.1 SCIENTIFIC FIGURE UNDERSTANDING TASKS

Significant research has been conducted in the domain of scientific document and figure analysis. Most prior works, however, have primarily focused on tasks such as figure classification (Jobin et al., 2019), figure detection (Roy et al., 2020), figure captioning (Hsu et al., 2021), and visual question answering and question generation over figures (Kahou et al., 2017). For instance, PDFMEF (Wu et al., 2015) proposed a multi-entity extraction framework designed to identify and extract figures from PDF documents. Similarly, ImageCLEF (Garcia Seco De Herrera et al., 2015) introduced a

benchmark dataset for compound figure detection and separation. PDFFigures (Clark & Divvala, 2015) facilitated the extraction of figures and their corresponding captions from research papers. PatentLMM (Shukla et al., 2025) generates high-quality descriptions of patent figures. Recently, DiagramAgent (Wei et al., 2024) extended this line of work by introducing a multi-agent framework for generating and editing diagrams from textual descriptions, supported by a diverse benchmark across multiple diagram types. In contrast, our work focuses on the task of scientific architecture diagram generation from natural language via intermediate DOT code, contributing a large-scale dataset with clean text-code-image alignment and a robust evaluation framework to advance semantic fidelity in figure understanding.

## 2.2 SCIENTIFIC FIGURES DATASETS

A range of classification-focused datasets, such as DocFigure (Jobin et al., 2019), FigureSteer (Siegel et al., 2016), Revision (Savva et al., 2011), and ACLFig (Karishma et al., 2023), support figure-type classification. FigureQA (Kahou et al., 2017) further introduced a large-scale dataset of over one million question-answer pairs grounded in synthetic figures such as bar graphs and line charts. However, none of these datasets are tailored to architecture or structured system diagrams.

To address this, SciCap (Hsu et al., 2021) introduced 60,000 figure-text pairs spanning diverse figure types, including flowcharts, equations, and plots. FigCap (Chen et al., 2019) similarly provided captions for a wide range of diagrams. Yet these datasets are highly heterogeneous and include many non-architectural figures, limiting their usefulness for architecture-specific generation. Automatikz (Belouadi et al., 2024a) introduced 120k TikZ drawings with captions, and DeTikZify (Belouadi et al., 2024b) expanded this to 360k. Paper2Fig (Rodriguez et al., 2023b) made a more focused effort to extract figures and context from research articles, but still contains many irrelevant figures (e.g., plots, tables) and noisy or incomplete text.

To overcome these limitations, we build upon Paper2Fig using extensive filtering and refinement. Specifically, we train image classifiers to retain only architecture-related diagrams and employ GPT-4o to generate clean, semantically rich descriptions. This curation produces a high-quality dataset focused exclusively on architecture diagrams with meaningful textual descriptions, suitable for text-to-architecture diagram generation.

## 3 TEXT2ARCH DATASET CURATION

In this section, we discuss our detailed dataset curation pipeline which is also illustrated in Fig. 2. The pipeline comprises 3 main steps: (1) Training Arch versus no-Arch Classifier, (2) DOT Code generation, and (3) Refining Architecture Image Descriptions.

**Training Arch versus no-Arch Classifier.** An image is classified as an architecture diagram if it visually represents the structural design, components, and relationships within a system, model, or process. This includes neural network architectures, software systems (diagrams illustrating microservices, databases, APIs, data pipelines, or system architecture), research figures (architecture figures in academic papers that describe model design, experimental setup, or algorithmic flow).

Although there exist multiple datasets with scientific diagrams, most of them do not have architecture diagrams specifically labeled. Hence, we train an arch vs no-arch classifier using a combination of three datasets obtained as follows. (1) ACL-Fig dataset (Karishma et al., 2023) has 1671 scientific figures (with 19 category labels) extracted from 890 research papers. Of these, we considered 103 neural networks and 105 architecture diagrams as positive. We considered remaining classes with 1474 images as negative. (2) SciFig dataset[4] does not have any labels, but has figure captions. Based on these captions, we identified 6482 images as positive examples. An equal number of images were randomly sampled from the remaining dataset to serve as negative examples. (3) Paper2Fig dataset contains 101371 images. We manually labeled 2004 images of which 1239 are positive images and 765 are negative. We keep 1003 of these for test purpose (620 arch and 383 no-arch).

Overall, we obtain a dataset of 7929 arch images and 8918 no-arch images. We use 1003 of Paper2Fig manually annotated subset for test purpose (620 arch and 383 no-arch). The remaining

---

[4]https://drive.google.com/drive/folders/1BNku_HcDPm3v4KKBj96u6X40IMieNo6D

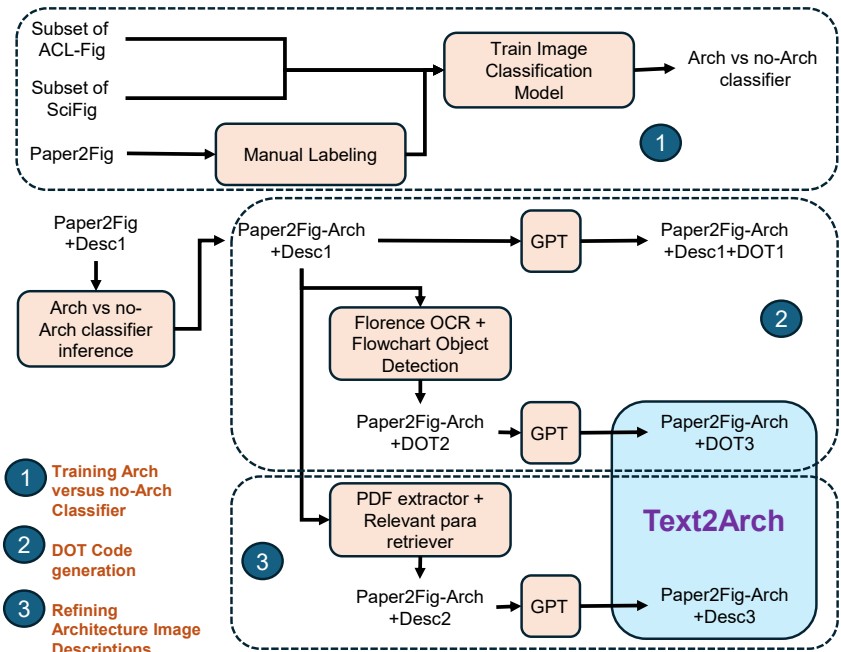

Figure 2: TEXT2ARCH Dataset Curation

images are stratified split into train and validation so as to maintain the same ratio of arch vs no-arch images in train and test.

We train multiple models like CLIP (Radford et al., 2021), ViT (Dosovitskiy et al., 2020), BEiT (Bao et al., 2021), and ResNet (He et al., 2016), and report results in Table 4 in Appendix A. We vary learning rate as 1e-5, 5e-4 and 5e-5. We train for up to 50 epochs and choose the best model based on validation loss. We also perform rotation, horizontal flip and vertical flip data augmentations.

Our best model is CLIP trained with a learning rate of 1e-5. It provides a test accuracy of 83.45% after just 2 epochs of training. The precision is 0.83, recall is 0.92 and F1 is 0.87. An accuracy of 83.45% is competitive given the inherent complexity and variability of scientific figures. More importantly, the high recall of 0.92 indicates that the classifier is effective at identifying architecture diagrams, which was our primary goal for downstream tasks. The F1 score of 0.87 further supports the model's balanced performance. Inferring this classifier over the entire Paper2Fig dataset gives us a set of 80486 architecture images.

**DOT Code Generation.** For training TEXT2ARCH models, our goal was to leverage the large Paper2Fig dataset. But it does not have any annotated DOT code. Hence, for each image in the dataset, we need to obtain ground truth DOT Code. Manual labeling is difficult for such a large set and hence we report to a combination of GPT-4o, florence2 OCR (Xiao et al., 2024) and Flowchart object detectors to obtain the final DOT code as follows.

First, we use a GPT prompt (Appendix E) which takes the figure as input and extracts the DOT code representation (called DOT1).

Next, we parse the architecture diagram using an object detection model from (Shukla et al., 2025). The model is trained on top of Faster-RCNN (Ren et al., 2015) using a dataset of 350 manually annotated patent figures and can extract diagram specific elements like nodes, node labels, figure labels, text and arrows. This model helps us extract nodes and edges for the DOT code. To extract the text representing the node, we use the florence2 OCR model (Xiao et al., 2024)[5]. Direction of arrows was decided based on detected arrow heads. Arrows were linked to nodes closest to start and end of arrow heads. To avoid self loops, we assign end of arrow to second nearest node if the nearest node is already associated with the arrow. We refer to this version of DOT code as DOT2. Further,

---

[5]microsoft/Florence-2-large-ft

we used GPT-4o to take DOT2+image as input and get refined DOT code (called DOT3). Detailed prompt is in Appendix G. Knowing the number of nodes in an image also helps us categorize images into problem complexity buckets: easy (upto 0-14 nodes), medium (15-24 nodes), hard (25+ nodes).

**Refining Architecture Image Descriptions.** As an input for TEXT2ARCH, we need a good image description. The Paper2Fig dataset already contains a paragraph (called Desc1) extracted from the paper containing the first reference of the figure. Besides this, the Paper2Fig dataset also contains a caption for each figure.

To obtain a more complete description, we first find all paragraphs from the paper text (extracted from the paper pdf using pypdf[6]) which contain the figure reference[7]. Next, we compute TF-IDF-based cosine similarity between all candidate paragraphs and a combination of OCR tags and image caption. We retain top 3 paragraphs with highest similarity score from this relevant paragraph retriever. We combine this with Desc1 and call it Desc2. Lastly, we use a GPT prompt to take the current image and these top 3 description paragraphs as input and output a revised description (Desc3). See detailed prompt in Appendix F.

**Overall TEXT2ARCH Dataset.** The Paper2Fig dataset subsetted to architecture diagram images along with DOT3 as labels and Desc3 as refined descriptions is further checked to remove entries where Desc3 or DOT3 is empty. We also remove entries where GPT labels images as no-Arch. The final filtered dataset is our contribution, referred to as TEXT2ARCH. We split the dataset into train, validation and test stratify by diagram complexity, i.e., the number of nodes. The dataset contains 60519 train samples, 7565 validation samples and 7043 test samples, adding to overall 75127 samples. Around 54.3% samples are easy, 32.4% are medium and the rest are of hard complexity.

Along with this dataset, a set of 99 images is manually labeled by the authors to obtain manually written DOT code. This dataset contains 50 easy, 30 medium and 19 hard images.

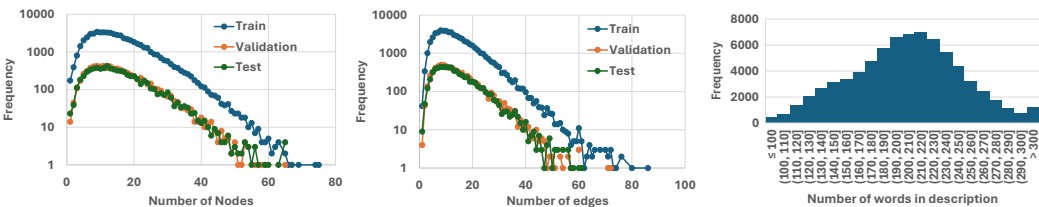

Figure 3: Distribution of number of nodes and edges, and number of words in image descriptions in TEXT2ARCH.

Fig. 3 shows the distribution of number of nodes and edges in the DOT graphs. The distributions are very similar across train, validation and test. On average there are 15.24 nodes and 13.89 edges per sample. The figure also shows the distribution of number of words in image descriptions in TEXT2ARCH. On average there are 203 words across these descriptions.

## 4 TEXT2ARCH METHODOLOGY

Given a clean input description about a system architecture, the goal of the proposed TEXT2ARCH system is to generate an architecture diagram. More specifically, in this work we focus on translating the natural language description to a DOT graph representing the architectural flow of processing in the input text. DOT compilers could then be leveraged to convert the DOT code to a graph. In this section, we discuss various models that we experiment with for TEXT2ARCH. We also propose novel task-specific metrics.

### 4.1 MODEL ARCHITECTURES AND APPROACHES

We benchmark several methods for the TEXT2ARCH task. First, we compare with the DiagramAgent method. Second, we compare with direct GPT-4o zero-shot inference. Third, we perform few-

---

[6]https://pypi.org/project/pypdf/

[7]Figure number preceded by any of these: Figure, Fig, figure, Figure., Fig., figure., fig., fig

shot inference with three popular instruction-tuned small language models. Lastly we also finetune three popular small language models.

For GPT-4o inference, we use the prompt as detailed in Appendix I. For few-shot inference, we perform inference with the following models: meta-llama/Meta-Llama-3-8B-Instruct (Dubey et al., 2024), Qwen/Qwen2-7B-Instruct (Qwen Team, 2024) and deepseek-ai/deepseek-llm-7b-chat (Liu et al., 2024a) using the prompt and the 5 few shot examples as listed in Appendix L. We perform few-shot prompting with instruction tuned models because we observed that the pretrained base versions failed to generate any reasonable output. With few shot prompting the pretrained models repeated the few shot examples in the output, or generated an empty output.

We selected Meta-Llama-3-8B-Instruct, Qwen2-7B-Instruct, and DeepSeek-LLM-7B-Chat for our experiments because they represent the current state-of-the-art among open-weight, instruction-tuned language models that balance performance, accessibility, and efficiency. These models are specifically fine-tuned for instruction-following and code generation tasks, aligning well with the requirements of generating structured DOT code from natural language descriptions. Compared to larger proprietary models, they are more lightweight and can be fine-tuned or deployed in resource-constrained environments. At the same time, they outperform earlier open models (like LLaMA-2 or GPT-J) on code-related benchmarks, and offer better controllability and interpretability than black-box diffusion-based text-to-image models. This combination of strong semantic understanding, structured output capabilities, and open accessibility makes them ideal candidates for our task.

As mentioned above, we perform supervised finetuning (SFT) of the base variants of these models (meta-llama/Meta-Llama-3-8B, Qwen/Qwen2-7B-base and DeepSeek-ai/DeepSeek-llm-7b) using this prompt: "You are an expert in analyzing technical descriptions of system architecture, workflows, and process pipelines, and a code design specialist skilled in graph visualization using DOT language."

| | | Text Metrics | | | | Graph Metrics | | | | | | | | |
|---|---|---|---|---|---|---|---|---|---|---|---|---|---|---|
| | | ROUGE-L | Code BLEU | Edit Distance | chrF | Node Prec | Node Recall | Node F1 | Node PR-AUC | Edge Prec | Edge Recall | Edge F1 | Edge PR-AUC | Jaccard Sim. |
| | DiagramAgent | 42.2 | 31.0 | 680 | 48.2 | 60.7 | 56.5 | 55.1 | 20.9 | 31.7 | 22.6 | 24.8 | 18.0 | 17.8 |
| | GPT | 30.8 | 17.7 | 730 | 42.9 | **71.6** | 56.5 | 60.7 | 27.4 | 56.3 | 39.5 | 44.6 | **31.6** | **36.1** |
| Few-shot ICL | Llama-3-8B | 34.9 | 21.5 | 709 | 41.0 | 69.6 | 56.8 | 59.7 | 24.2 | 41.5 | 29.0 | 32.5 | 22.0 | 24.6 |
| Few-shot ICL | Qwen2-7B | 27.4 | 19.4 | 811 | 30.4 | 64.9 | 48.1 | 52.0 | 19.8 | 32.8 | 21.2 | 24.3 | 15.9 | 17.2 |
| Few-shot ICL | DeepSeek-7B | 30.4 | 21.4 | 1079 | 31.4 | 54.1 | 41.3 | 43.5 | 17.4 | 25.0 | 13.6 | 16.6 | 13.0 | 11.5 |
| Fine-tuned | Llama-3-8B | 28.2 | 27.8 | 956 | 39.8 | 27.8 | 45.3 | 31.9 | 7.0 | 22.9 | 10.2 | 13.2 | 9.1 | 8.4 |
| Fine-tuned | Qwen2-7B | 35.0 | 30.7 | 975 | 45.3 | 33.3 | 49.8 | 36.8 | 8.1 | 21.8 | 10.8 | 13.7 | 8.8 | 8.8 |
| Fine-tuned | DeepSeek-7B | **46.8** | **34.5** | **608** | **55.7** | 66.2 | **69.6** | **65.7** | 21.5 | 46.4 | 34.2 | 38.0 | 23.7 | 28.6 |

Table 1: DOT code generation task results on TEXT2ARCH test set. (Best values are in bold. Second best are underlined.)

| | | Text Metrics | | | | Graph Metrics | | | | | | | | |
|---|---|---|---|---|---|---|---|---|---|---|---|---|---|---|
| | | ROUGE-L | Code BLEU | Edit Distance | chrF | Node Prec | Node Recall | Node F1 | Node PR-AUC | Edge Prec | Edge Recall | Edge F1 | Edge PR-AUC | Jaccard Sim. |
| | DiagramAgent | 49.1 | 40.9 | 959 | 55.3 | 55.0 | 59.8 | 54.3 | 12.4 | 32.0 | 22.8 | 25.3 | 16.9 | 17.8 |
| | GPT | 28.2 | 16.3 | 790 | 43.4 | **69.7** | 60.1 | 63.0 | **28.0** | 55.9 | 40.4 | 46.2 | 30.9 | 37.8 |
| Few-shot ICL | Llama-3-8B | 37.3 | 23.1 | 474 | 43.6 | 62.4 | 52.9 | 54.1 | 17.0 | 42.4 | 28.5 | 32.5 | 17.4 | 25.7 |
| Few-shot ICL | Qwen2-7B | 30.1 | 22.0 | 562 | 32.0 | 54.9 | 50.7 | 48.9 | 13.2 | 37.1 | 21.7 | 25.5 | 14.9 | 18.1 |
| Few-shot ICL | DeepSeek-7B | 36.9 | 28.6 | 872 | 35.8 | 52.1 | 42.2 | 42.9 | 13.8 | 27.5 | 17.5 | 20.4 | 13.9 | 15.4 |
| Fine-tuned | Llama-3-8B | 30.9 | 46.0 | 1024 | 44.0 | 21.3 | 46.4 | 27.5 | 4.9 | 27.7 | 11.9 | 15.2 | 14.3 | 10.1 |
| Fine-tuned | Qwen2-7B | 40.9 | 46.3 | 891 | 53.1 | 35.3 | 57.1 | 40.2 | 9.4 | 19.4 | 11.5 | 14.0 | 11.8 | 9.3 |
| Fine-tuned | DeepSeek-7B | **55.2** | **49.3** | **407** | **66.6** | 66.1 | **78.1** | **69.4** | 27.4 | **59.4** | **44.6** | **49.1** | **35.1** | **39.8** |

Table 2: DOT code generation task results on TEXT2ARCH manual annotation set. (Best values are in bold. Second best are underlined.)

## 4.2 METRICS

For evaluating the TEXT2ARCH task, we assess the generated DOT code using the following standard natural language generation (NLG) metrics as well as the novel graph-based metrics which we design specifically for the TEXT2ARCH task. The standard NLG metrics include the following. (1) **ROUGE-L** (↑): Measures similarity with the reference code based on the Longest Common Subsequence (LCS), focusing on recall. (2) **CodeBLEU** (↑): Assesses the semantic and syntactic sim-

ilarity with reference code, incorporating n-gram matching, Abstract Syntax Tree (AST) matching, and data-flow matching. (3) **Edit Distance (↓):** Computes the Levenshtein distance, representing the minimum number of single-character edits (insertions, deletions, or substitutions) required to change the generated code into the reference code. (4) **chrF (↑):** Character n-gram F-score, evaluates text similarity at the character level, making it robust to tokenization differences.

The graph-based evaluation metrics evaluate the structural correctness of the generated diagrams by comparing the underlying graph structures derived from the DOT code. We define the following graph-based metrics suitable to this task. (1) **Node Precision, Recall, F1-Score (↑):** These metrics assess the accuracy of node identification and labeling. Node matching is performed using string similarity between labels (via Hungarian algorithm for optimal assignment), and scores are aggregated. (2) **Node PR-AUC (Area Under Precision-Recall Curve) (↑):** Measures the overall performance of node matching across various similarity thresholds. (3) **Edge Precision, Recall (↑):** Evaluate the correctness of detected edges between matched nodes. (4) **Edge PR-AUC (Area Under Precision-Recall Curve) (↑):** Measures the overall performance of edges across various similarity thresholds. (5) **Jaccard Similarity (Edges) (↑):** Measures the overlap between the sets of edges in the ground truth and predicted graphs, based on matched nodes. Detailed definitions of these metrics are included in Appendix C.

| | Text Metrics | | | | Graph Metrics | | | | | | | | |
|---|---|---|---|---|---|---|---|---|---|---|---|---|---|
| | ROUGE-L | Code BLEU | Edit Distance | chrF | Node Prec | Node Recall | Node F1 | Node PR-AUC | Edge Prec | Edge Recall | Edge F1 | Edge PR-AUC | Jaccard Sim. |
| DOT1 | 13.8 | 35.3 | 1843 | 37.1 | 66.1 | 78.4 | 67.5 | 28.5 | 44.6 | 25.8 | 31.2 | 30.2 | 22.9 |
| DOT2 | 29.1 | 40.4 | **422** | 58.7 | 52.9 | 62.0 | 54.6 | 13.4 | 15.6 | 6.2 | 8.3 | 6.4 | 5.1 |
| DOT3 | **56.2** | **49.4** | 620 | **68.0** | **71.0** | **83.5** | **74.5** | **33.7** | **63.0** | **46.6** | **51.7** | **39.5** | **41.2** |

Table 3: Comparison of DOT code variants as judged on TEXT2ARCH manual annotation set. Best values are in bold.

## 5 EXPERIMENTS

### 5.1 MAIN RESULTS

We first list our main results using automated metrics, followed by a GPT-based evaluation. we also discuss quality comparison across various DOT and Desc variants. Lastly, we show several examples of generated output from TEXT2ARCH and other models in the Appendix.

Tables 1 and 2 summarize the performance of various methods on the text-to-DOT code generation task, evaluated on both the TEXT2ARCH test set and the manually annotated set. Note that for computation of these metrics, we used DOT3 variant from TEXT2ARCH dataset as the ground truth. Note that DiagramAgent generates TikZ code rather than DOT code. Hence, we use the prompt detailed in Appendix H with GPT-4o to convert the generated TikZ code to DOT.

The results clearly demonstrate the effectiveness of finetuned models over few-shot in-context learning (ICL) models and GPT. On text-based metrics such as ROUGE-L, CodeBLEU, Edit Distance, and chrF, finetuned models, particularly the DeepSeek-7B, achieve substantial improvements. For example, on the manual annotation set, DeepSeek-7B achieves a ROUGE-L of 55.2 and CodeBLEU of 49.3, significantly outperforming the best few-shot ICL model (Llama-3-8B, ROUGE-L 37.3, CodeBLEU 23.1) and GPT (ROUGE-L 28.2, CodeBLEU 16.3). This indicates that finetuning enables the models to generate code that is not only more syntactically and semantically aligned with the ground truth but also more concise and accurate, as reflected in lower edit distances and higher chrF scores.

Beyond text similarity, the graph-based metrics provide deeper insights into the structural fidelity of the generated diagrams. Finetuned models and GPT perform better than few-shot ICL models, with DeepSeek-7B achieving the highest node and edge F1 scores (e.g., Node F1 65.7/69.4 and Edge F1 38.0/49.1 on the test/manual sets, respectively). These gains are also reflected in higher PR-AUC and Jaccard similarity scores, underscoring that finetuning improves not just the surface-level code but also the underlying DOT graph semantics. Performance from few-shot ICL models seems inconsistent: Llama-3-8B and Qwen2-7B few-shot models are better but DeepSeek-7B few-shot models are worse compared to their finetuned counterparts.

## 5.2 GPT-4O BASED EVALUATION

Besides automated metric based evaluation, we also perform GPT based evaluation to compare the generated DOT code with both the image and the ground-truth DOT code. We ask GPT to determine if the structure, labels, node ordering, and relationships in the generated DOT code accurately reflect the image and ground-truth. And then assign a compatibility score between 0 and 5. Detailed prompt is in Appendix K.

GPT-4o (2.72) and DeepSeek-7B (2.68) outperformed both the DiagramAgent baseline (2.37) and the other 7B-scale models (Qwen2-7B at 2.24, Llama-3-8B at 1.90). This subjective assessment mirrors the objective gains we see in Table 1: on text metrics (ROUGE-L, CodeBLEU, chrF) DeepSeek-7B consistently leads the finetuned group (e.g. ROUGE-L 46.8 vs. 30.8 for GPT, 42.2 for Diagram-Agent), and on graph metrics it achieves the highest node/edge F1 and PR-AUC scores.

In short, compared to other small language models and DiagramAgent baseline, the DeepSeek-7B model not only "looks" better to an expert evaluator but also delivers the best syntactic accuracy and structural fidelity across both tables, validating that human judgments align closely with our quantitative benchmarks.

## 5.3 QUALITY COMPARISON BETWEEN VARIOUS DOT VARIANTS

Table 3 highlights the significant improvements achieved by DOT3 over variants DOT1 and DOT2. DOT3 achieves the highest scores across both text-based and graph-based metrics, with ROUGE-L 56.2, CodeBLEU 49.4, and chrF 68.0, indicating much closer alignment to the ground truth code. On structural metrics, DOT3 shows substantial gains in both node and edge quality, achieving Node F1 74.5 and Edge F1 51.7, compared to DOT1 (Node F1 67.5, Edge F1 31.2) and DOT2 (Node F1 54.6, Edge F1 8.3). These results demonstrate that DOT3 not only preserves the semantic content of the diagrams better than DOT1, but also corrects the structural inconsistencies and low edge quality observed in DOT2. The high Jaccard similarity of 41.2 further confirms that DOT3 captures overall graph topology more faithfully.

These improvements validate the design of our multi-step pipeline for generating high-quality DOT code. DOT1, derived directly from GPT prompting, suffers from incomplete or noisy outputs due to the inherent limitations of language models in precise spatial reasoning. DOT2, constructed from object detection and OCR, improves node and text alignment but struggles with edge connectivity and overall coherence, as reflected in its low edge metrics. By combining the structural grounding of DOT2 with GPT-based refinement in DOT3, we effectively leverage the strengths of both approaches, accurate detection of components and the generative ability of GPT to produce syntactically correct and semantically coherent DOT code. This hybrid strategy proves crucial for producing high-fidelity representations of complex diagrams, as evidenced by the consistent gains across all evaluation metrics.

## 5.4 QUALITY COMPARISON BETWEEN VARIOUS DESCRIPTION VARIANTS

Using GPT, we evaluate whether Desc1 or Desc3 is a better description for an image. Similarly, we evaluate between Desc2 and Desc3. The prompt is detailed in Appendix J. We also experimented with positions exchanged in the prompt to eliminate position bias. We found that Desc3 was preferred over both Desc1 and Desc2 over 90% times.

## 5.5 QUALITATIVE ANALYSIS

We show 3 case studies comparing outputs from various models for the TEXT2ARCH task in Tables 4 to 8 in Appendix D. Our TEXT2ARCH's finetuned DeepSeek-7B consistently demonstrates superior performance in structured diagram synthesis compared to all baseline models. Whether reconstructing neural architectures, algorithmic pipelines, or domain-specific workflows, DeepSeek-7B excels in both semantic fidelity and structural accuracy. It reliably captures node labels, edge relationships, and even nuanced features like skip connections and module-specific operations. In contrast, baselines such as DiagramAgent, GPT, and few-shot DeepSeek variants often produce incomplete, generic, or misaligned outputs. The ground truth DOT codes, when available, are frequently noisy or under-specified, further highlighting the clarity and precision of DeepSeek-7B's

outputs. These results underscore the value of fine-tuning for domain-specific tasks and position DeepSeek-7B as a robust solution for automated diagram generation in technical and scientific contexts.

## 6 CONCLUSION

In this work, we introduced the novel task of generating scientific architecture diagrams from natural language descriptions via intermediate DOT code, addressing a critical gap in structured diagram generation. We contributed TEXT2ARCH, a large-scale, high-quality dataset of over 75K aligned text–code–image triplets, enabling rigorous benchmarking of this task. Through extensive experiments, we demonstrated that fine-tuned small language models significantly outperform both few-shot in-context learning and existing baselines like DiagramAgent, achieving superior syntactic accuracy and structural fidelity. Our proposed evaluation framework, combining text-level and graph-level metrics, provides a comprehensive assessment of both semantic and structural correctness. Notably, the fine-tuned DeepSeek-7B model emerged as the most effective, closely matching GPT-4o in evaluations on the human-generated test set while being open and lightweight. These findings highlight the importance of domain-specific fine-tuning and structured evaluation in advancing text-to-architecture generation. We make the code, data and models publicly available[1]. We believe our dataset, models, and insights will catalyze further research in this emerging area.

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

# Overview of Appendix Sections

- Appendix A: Training Arch versus no-Arch Classifier Detailed Results
- Appendix B: Hyper-parameters for reproducibility
- Appendix C: Detailed Definitions of Evaluation Metrics
- Appendix D: Case Studies
- Appendix E: GPT prompt to obtain DOT code representation from architecture figure
- Appendix F: GPT prompt to get a refined image description
- Appendix G: GPT prompt to refine DOT code
- Appendix H: GPT Prompt to convert TikZ code to DOT
- Appendix I: Zero-shot GPT prompt to obtain DOT code from architecture description
- Appendix J: GPT Prompt to Compare Descriptions
- Appendix K: GPT prompt for TEXT2ARCH task evaluation
- Appendix L: Few Shot Examples for the Small Language Models based evaluation
- Appendix M: Detailed Description of Model Architectures
- Appendix N: Prompts and Results for Lengthened and Shortened Descriptions
- Appendix O: Human Evaluation

## A  TRAINING ARCH VERSUS NO-ARCH CLASSIFIER DETAILED RESULTS

We train multiple models like CLIP (Radford et al., 2021), ViT (Dosovitskiy et al., 2020), BEiT (Bao et al., 2021), ResNet (He et al., 2016) and report results in Table 4. We vary learning rate as 1e-5, 5e-4 and 5e-5. We train for up to 50 epochs and choose the best model based on validation loss. We also perform rotation, horizontal flip and vertical flip data augmentations.

Our best model is CLIP trained with a learning rate of 1e-5. It provides a test accuracy of 83.45% after just 2 epochs of training. The precision is 0.83, recall is 0.92 and F1 is 0.87. Inferring this classifier over the entire Paper2Fig dataset gives us a set of 80486 architecture images.

## B  HYPER-PARAMETERS FOR REPRODUCIBILITY

All experiments were run on a machine with 8 NVIDIA V100 GPUs.

For training arch vs no-arch classifiers, we used batch size of 64, learning rate of 5e-5, and AdamW optimizer.

For supervised full finetuning of the TEXT2ARCH models, we trained using AdamW optimizer with batch size of 1 per device, gradient accumulation steps of 4, learning rate of 5e-4, weight decay of 0.001, max gradient norm of 0.3, warmup ratio of 0.03, cosine learning rate scheduler, and 5 epochs with DeepSpeed ZeRO3.

For model inferences, we used temperature of 0.7, top_p of 0.9, and max_length of 1024. For GPT-4o inferences, we used temperature of 0.15, max_tokens of 1000, top_p of 0.8, frequency_penalty of 1, and presence_penalty of 1.

For evaluation metrics, we used a similarity threshold of 0.5 for node matching in graph metrics calculations.

Node matching details: We use SequenceMatcher from difflib for computing string similarity. We perform basic normalization by converting labels to lowercase and removing extra whitespace and newlines. We compute both a character-level similarity using SequenceMatcher and a token-level Jaccard similarity, taking the maximum of the two.

DOT parsing/canonicalization: Subgraphs are parsed as-is using networkx.drawing.nx_agraph.read_dot. Multi-edges and duplicate edges are not explicitly handled or deduplicated. Self-loops are preserved. Rank and positional attributes (e.g., rank=same, pos, layout hints) are ignored, as the evaluation focuses on structural and label-based similarity rather than layout fidelity. Edge directionality is preserved. All edge-based metrics (precision, recall, Jaccard) assume directed edges.

| Model | Learning Rate | Epoch | Val Loss | Val Acc | Test Acc |
|-------|---------------|-------|----------|---------|----------|
| BEiT | 1.00E-05 | 33 | 0.9875 | 86.2 | 0.7886 |
| BEiT | 5.00E-04 | 19 | 0.5973 | 71.3 | 0.657 |
| BEiT | 5.00E-05 | 44 | 1.1392 | 84.45 | 0.7906 |
| CLIP | 1.00E-05 | 2 | 0.3389 | 87.55 | 0.8345 |
| CLIP | 5.00E-04 | 36 | 0.6564 | 62.9 | 0.5533 |
| CLIP | 5.00E-05 | 0 | 0.3978 | 83.95 | 0.7647 |
| ResNet | 1.00E-05 | 27 | 0.3944 | 85.03 | 0.7936 |
| ResNet | 5.00E-04 | 45 | 1.1516 | 86.05 | 0.8036 |
| ResNet | 5.00E-05 | 4 | 0.3774 | 86.81 | 0.7966 |
| ViT | 1.00E-05 | 1 | 0.3797 | 85.08 | 0.8185 |
| ViT | 5.00E-04 | 4 | 0.5684 | 77.6 | 0.6849 |
| ViT | 5.00E-05 | 10 | 0.6373 | 86.46 | 0.7787 |

Table 4: Architecture-image versus No-Architecture-Image Classification Results

# C   DETAILED DEFINITIONS OF EVALUATION METRICS

## C.1   STANDARD NLG METRICS

### C.1.1   ROUGE-L

ROUGE-L (Recall-Oriented Understudy for Gisting Evaluation - Longest Common Subsequence) measures the similarity based on the length of the longest common subsequence (LCS) between the predicted code $C_{pred}$ and the reference code $C_{ref}$. Let $LCS(C_{pred}, C_{ref})$ be the length of the longest common subsequence. Let $m = \text{length}(C_{pred})$ and $n = \text{length}(C_{ref})$.

$$R_{\text{LCS}} = \frac{LCS(C_{pred}, C_{ref})}{n}$$

$$P_{\text{LCS}} = \frac{LCS(C_{pred}, C_{ref})}{m}$$

$$\text{ROUGE-L} = \frac{(1 + \beta^2) R_{\text{LCS}} P_{\text{LCS}}}{R_{\text{LCS}} + \beta^2 P_{\text{LCS}}}$$

Typically, $\beta$ is set to a large value to emphasize recall, or F-measure is reported where $\beta = 1$.

### C.1.2   CODEBLEU

CodeBLEU is a composite score that evaluates code generation quality by considering n-gram match (BLEU), weighted n-gram match, Abstract Syntax Tree (AST) match, and data-flow match. The final score is a weighted combination of these components.

$$\text{CodeBLEU} = w_1 \cdot \text{BLEU}_{\text{ngram}} + w_2 \cdot \text{BLEU}_{\text{weighted}}$$
$$+ w_3 \cdot \text{Match}_{\text{AST}} + w_4 \cdot \text{Match}_{\text{dataflow}}$$

where $w_i$ are the weights for each component.

### C.1.3   EDIT DISTANCE

Edit Distance, specifically Levenshtein distance, is the minimum number of single-character edits (insertions, deletions, or substitutions) required to change the predicted code $C_{pred}$ into the reference code $C_{ref}$. It is denoted as $Lev(C_{pred}, C_{ref})$.

### C.1.4   CHRF

chrF (character n-gram F-score) computes the F-score based on overlapping character n-grams between the predicted code $C_{pred}$ and reference code $C_{ref}$. It is less sensitive to tokenization issues than word-based metrics. Let $chrP_n$ and $chrR_n$ be the precision and recall for character n-grams of length $n$. The chrF score is typically a weighted average over different n-gram lengths (e.g., 1 to 6).

## C.2   GRAPH-BASED METRICS

These metrics operate on the graph representations $G_{gt} = (V_{gt}, E_{gt})$ (ground truth) and $G_{pred} = (V_{pred}, E_{pred})$ (predicted). Node matching is performed first. Let $M$ be the set of matched pairs $(v_{gt}, v_{pred})$ where $v_{gt} \in V_{gt}$ and $v_{pred} \in V_{pred}$, typically found using string similarity of labels and the Hungarian algorithm for optimal assignment above a certain similarity threshold $\tau$.

### C.2.1   NODE PRECISION, RECALL, F1-SCORE

Let $N_{matched} = |M|$ be the number of matched nodes. Let $N_{gt} = |V_{gt}|$ and $N_{pred} = |V_{pred}|$. The script 'orig_metric_calc.py' calculates similarity-weighted precision and recall. For a match $(v_{gt}, v_{pred})$ with similarity $sim(v_{gt}, v_{pred})$:

$$\text{Node Precision} = \frac{\sum_{(v_{gt}, v_{pred}) \in M} sim(v_{gt}, v_{pred})}{N_{pred}}$$

$$\text{Node Recall} = \frac{\sum_{(v_{gt}, v_{pred}) \in M} sim(v_{gt}, v_{pred})}{N_{gt}}$$

$$\text{Node F1} = \frac{2 \cdot \text{Node Precision} \cdot \text{Node Recall}}{\text{Node Precision} + \text{Node Recall}}$$

### C.2.2 NODE PR-AUC

This is the Area Under the Precision-Recall curve, obtained by varying the string similarity threshold $\tau$ for node matching and plotting the resulting (Node Recall, Node Precision) pairs.

### C.2.3 EDGE PRECISION, RECALL

Based on the set of matched nodes $M$, we consider edges. Let $E_{gt\_matched}$ be the set of edges in $G_{gt}$ between nodes that have a match in $M$. Similarly for $E_{pred\_matched}$. An edge $(u_{gt}, v_{gt})$ in $G_{gt}$ is considered correctly predicted if there is a corresponding edge $(u_{pred}, v_{pred})$ in $G_{pred}$ where $(u_{gt}, u_{pred}) \in M$ and $(v_{gt}, v_{pred}) \in M$. Let $TP_E$ be the number of such correctly predicted edges. Let $FP_E$ be the number of edges in $G_{pred}$ between matched nodes that do not correspond to an edge in $G_{gt}$. Let $FN_E$ be the number of edges in $G_{gt}$ between matched nodes that do not have a corresponding edge in $G_{pred}$.

$$\text{Edge Precision} = \frac{TP_E}{TP_E + FP_E}$$

$$\text{Edge Recall} = \frac{TP_E}{TP_E + FN_E}$$

### C.2.4 EDGE PR-AUC

This is the Area Under the Precision-Recall curve, obtained by varying the string similarity threshold $\tau$ for node matching and plotting the resulting (Edge Recall, Edge Precision) pairs.

### C.2.5 JACCARD SIMILARITY (EDGES)

This considers the set of edges present in the adjacency matrices of the matched subgraphs. Let $Adj_{gt}$ be the adjacency matrix for $G_{gt}$ considering only nodes in $M_1 = \{v_{gt} | (v_{gt}, v_{pred}) \in M\}$. Let $Adj_{pred}$ be the adjacency matrix for $G_{pred}$ considering only nodes in $M_2 = \{v_{pred} | (v_{gt}, v_{pred}) \in M\}$.

$$\text{Jaccard Index} = \frac{|\text{Edges}(Adj_{gt}) \cap \text{Edges}(Adj_{pred})|}{|\text{Edges}(Adj_{gt}) \cup \text{Edges}(Adj_{pred})|}$$

This simplifies to $\frac{TP_E}{TP_E + FP_E + FN_E}$.

## D CASE STUDIES

We show 3 case studies comparing outputs from various models for the TEXT2ARCH task in Tables 4 to 8.

Case Study 1 (Table 4): The finetuned DeepSeek-7B model significantly outperforms all baselines, including DiagramAgent, GPT, and few-shot prompted DeepSeek variants, in terms of node and edge fidelity. Its output closely mirrors the original diagram, accurately capturing the flow from input signals through synaptic weights, summing junction, activation function, and final output. In contrast, baseline models produce less precise or overly generic representations. This highlights the effectiveness of fine-tuning for structured diagram synthesis and the potential of DeepSeek-7B in tasks requiring high semantic and structural accuracy.

Case Study 2 (Table 6): This case study analyzes the performance of various models in generating DOT representations of the TRIM algorithm pipeline, which accelerates image registration through coarse triangulation. The original diagram outlines a clear, linear sequence of six modules, each with distinct semantic roles. The finetuned DeepSeek-7B model demonstrates perfect structural and semantic fidelity, accurately capturing both the node labels and the directed flow of operations.

In contrast, the DiagramAgent baseline introduces incorrect edges and misordered steps, while the few-shot DeepSeek and GPT models show partial correctness but miss key relationships or introduce redundant paths. The ground truth DOT code is incomplete and inconsistent, further emphasizing the superior performance of DeepSeek-7B. This analysis highlights the model's ability to understand and reconstruct domain-specific pipelines with high precision, making it a strong candidate for automated diagram synthesis in technical documentation.

Case Study 3 (Table 8): This case study evaluates the ability of different models to reconstruct the architecture of a neural network designed for iris recognition, as depicted in the original figure. The network includes a sequence of convolutional and pooling layers, culminating in a softmax output, with a notable skip connection from the average pooling layer to a later convolutional layer. The finetuned DeepSeek-7B model demonstrates high fidelity to the original structure, accurately capturing both the layer sequence and the skip connection using a dashed edge. In contrast, the DiagramAgent baseline introduces redundant edges and misrepresents the skip connection, while the few-shot DeepSeek and GPT models show partial correctness but either omit or misplace key connections. The ground truth DOT code is noisy and inconsistent, further highlighting the clarity and precision of DeepSeek-7B's output. This analysis reinforces the model's strength in understanding and reproducing complex neural architectures with structural and semantic accuracy.

We also show side-by-side visual diagrams generated by all evaluated models (DiagramAgent, GPT, and our few-shot prompting based DeepSeek model) for the three case studies in Figs. 5, 7 and 9 respectively. The improvements obtained using our model are very clear from these illustrations. Baseline methods lead to several errors like structural mismatches, missing nodes, incorrect edges, and rendering artifacts.

# E   GPT PROMPT TO OBTAIN DOT CODE REPRESENTATION FROM ARCHITECTURE FIGURE

You are an expert **in** analyzing images, doing OCR and extracting structured flowchart information from images. I am trying to parse this architecture diagram into a text−based graph file. First give me OCR output **for** this image. Can you please give me DOT code **for** this image?
IMAGE:*#url#*

Output all of these **in** a nested XML. OCR output should be within <ocr></ocr> tags, DOT output should be within <dot></dot> tags. The ocr and DOT tags should be within <results></results> tags.

# F   GPT PROMPT TO GET A REFINED IMAGE DESCRIPTION

You are an expert **in** analyzing research materials, particularly visual content like architecture diagrams. Your task is to examine an input image alongside up to 3 descriptive paragraphs and a caption that are relevant to the image. These paragraphs may include a mix of relevant details and extraneous information about the image.

Your goal is to:
1. Determine **if** the image is an architecture diagram (commonly used **in** research papers to depict the structure, components, or workflows of systems).
2. If it is an architecture diagram, generate a concise, precise, and coherent description of 10−20 sentences explaining the main elements of the diagram. Description should include module names, short description of modules, and flow of information across modules.
3. The description you provide would be further used to train a model to generate such architecture images. Hence avoid any irrelevant information **in** the description.

*## Requirements:*
1. Carefully analyze the provided paragraphs, focusing on extracting key elements that directly explain the architecture depicted **in** the image.

2. Exclude extraneous, redundant, or noisy details from the textual content and focus only on the architectural aspects.
3. Clearly indicate whether the image is an architecture diagram or not.
4. Provide your output **in** a structured format

*## Inputs:*
IMAGE:*#imageURL#*
Caption: *#caption#*
Description: *#Descriptions#*

*## Example Output:*

<results>
    <label>[arch|not arch]</label>
    <newDesc>Concise and precise description goes here.</newDesc>
</results>

Output results **in** a nested XML. Label output should be within <label></label> tags and could be "arch" or "not arch". A brief, clear description of the image based on your understanding of the image and provided passages and caption should be within <newDesc></newDesc> tags. The label and newDesc tags should be within <results></results> tags.

# G    GPT PROMPT TO REFINE DOT CODE

You are an expert **in** analyzing research materials, particularly visual content such as architecture diagrams and system diagrams, and a code design specialist.

Your task is to:

1. Analyze both the DOT code and the image to identify any incorrect node labels, incorrect connections, or incorrect ordering of nodes.
2. Refine the DOT code to ensure it accurately represents the structure and relationships depicted **in** the image.
3. Output the corrected DOT file **in** a structured XML format.

*## Inputs:*
IMAGE:*#imageURL#*
Initial DOT code: (which may contain errors or incomplete data)
*#dotCode#*

*## Example Output:*
```
<results>
    <![CDATA[
        digraph {
            0 [label=''Node 0 description'']
            1 [label=''Node 1 description'']
            2 [label=''Node 2 description'']
            3 [label=''Node 3 description'']
            0 -> 1;
            0 -> 2;
            2 -> 3;
        }
    ]]>
</results>
```

*## Instructions:*
1. Ensure the refined DOT code fully represents the relationships **in** the image.
2. Maintain proper indentation and formatting **in** the DOT code.
3. Encapsulate the final DOT code within <results><![CDATA[ ]]></results> to prevent XML parsing issues.

# H    GPT PROMPT TO CONVERT TIKZ CODE TO DOT

Given the following LaTeX TikZ code:

1. First, re−indent the TikZ part **for** readability.

2. Then, extract all \node text labels and assign each a unique integer ID (e.g., 0, 1, 2...).

3. Use the format: ID [label=''...'']; to define each node.

4. Infer reasonable directed edges based on layout or label semantics (e.g., data flow, left−to−right, top−to−bottom).

5. Output the result as a DOT file using the below graph structure, starting directly with:

```
<results>
    <![CDATA[
        digraph {
            0 [label=''Node 0 description'']
            1 [label=''Node 1 description'']
            2 [label=''Node 2 description'']
            3 [label=''Node 3 description'']
            0 −> 1;
            0 −> 2;
            2 −> 3;
        }
    ]]>
</results>
```

Do not include any rankdir or node settings.
Maintain proper indentation and formatting **in** the DOT code.
Encapsulate the final DOT code within <results><![CDATA[ ]]></results> to prevent XML parsing issues.

Here is the TikZ code:
*#TikZCode#*

# I    ZERO-SHOT GPT PROMPT TO OBTAIN DOT CODE FROM ARCHITECTURE DESCRIPTION

You are an expert **in** analyzing technical descriptions of system architecture, workflows, and process pipelines, and a code design specialist skilled **in** graph visualization using DOT language.

Your task is to:
1. Read and interpret the following textual description of a system, pipeline, or process.
2. Generate accurate DOT code that reflects the described structure, relationships, and flow.
3. Output the DOT code **in** a structured XML format **for** downstream usage.

*## Input:*
*#Cleaned−Description#*

*## Example Output:*
```
<results>
    <![CDATA[
        digraph {
            0 [label=''Node 0 description'']
            1 [label=''Node 1 description'']
            2 [label=''Node 2 description'']
            3 [label=''Node 3 description'']
            0 −> 1;
            0 −> 2;
```

```
            2 -> 3;
        }
    ]]>
</results>
```

Instructions:
– Identify all relevant entities (nodes) and their relationships (edges) from the input description.
– Use DOT digraph syntax to define the flow or structure.
– Ensure that node relationships accurately reflect direction and logic described **in** the text.
– Format the DOT code cleanly and consistently with appropriate indentation.
– Wrap the DOT code inside a CDATA section within XML to avoid escaping issues.
– Do not provide explanations or additional commentary; only **return** the XML block containing the DOT code.

## J  GPT PROMPT TO COMPARE DESCRIPTIONS

Your task is to:
1. Analyze the given image along with two candidate textual descriptions (marked as Description 1 and Description 2).
2. Determine which description better matches the content and semantics of the image.
3. Return the index of the better matching description (either 1 or 2), followed by a short explanation justifying your choice.

*## Inputs:*
Image:
IMAGE:*#Image_URL#*
Description 1: *#description_1#*
Description 2: *#description_2#*

*## Output Format:*
Output all of these **in** a nested XML.
```
<results>
    <index>1</index>
    <explanation>The explanation should briefly describe why the selected description matches the image better.</explanation>
</results>
```

*## Evaluation Criteria:*
– Accuracy of objects, people, and actions mentioned **in** the description.
– Correctness of spatial relationships or layout depicted.
– Relevance of the description to the overall theme and content of the image.

## K  GPT PROMPT FOR TEXT2ARCH TASK EVALUATION

You are an expert **in** visual content analysis and structured graph representations.

Your task is to evaluate how well a generated DOT code matches a given ground–truth DOT code and the corresponding reference image.

*## Inputs:*
– Image:
IMAGE:*#Image_URL#*
– Generated DOT code: *#Generated–DOT#*
– Ground–truth DOT code: *#Ground–Truth–DOT#*

*## Instructions:*
1. Analyze the image to understand the correct structure, node positions, labels, and connections.
2. Compare the generated DOT code with both the image and the ground–truth DOT code.

3. Determine **if** the structure, labels, node ordering, and relationships **in** the generated DOT code accurately reflect the image and ground−truth.
4. Assign a compatibility score between 0 and 5, where:
    − 5 = Perfect match.
    − 4 = Minor discrepancies that don't affect comprehension.
    − 3 = Some noticeable errors, but mostly accurate.
    − 2 = Multiple mismatches that affect comprehension.
    − 1 = Mostly incorrect.
    − 0 = Completely unrelated.
5. Provide a concise explanation (2−3 sentences) describing the key issues or strengths.

## Output Format:

<results>
    <score>4</score>
    <explanation>The generated DOT code has correct node labels and most connections, but the order and direction of two edges differ from the image.</explanation>
</results>

Output all of these in a nested XML.

## L  FEW-SHOT PROMPT FOR INSTRUCT MODEL INFERENCE

We use the following few-shot prompt.

You are an expert **in** analyzing technical descriptions of system architecture, workflows, and process pipelines, and a code design specialist skilled **in** graph visualization using DOT language.

Your task is to convert technical descriptions into DOT graph representations. Follow these guidelines:

1. Use 'digraph {' as the graph declaration.
2. Set appropriate rankdir (TB for top−bottom, LR for left−right) if needed
3. Use appropriate node shapes (box is default)
4. Create meaningful node labels
5. Add edge labels where appropriate to describe relationships
6. Keep the graph structure clear and readable
7. IMPORTANT: Respond with ONLY the DOT code, no explanations or additional text

Here are examples of how to convert descriptions to DOT graphs: {few_shot_examples}.

Convert the following description into DOT language code. Respond with ONLY the DOT code and nothing else: {description}''.

### Few Shot Examples for the Small Language Models based evaluation

**Example 1**  Description: A simple web application architecture with a frontend that connects to a backend API, which then connects to a database.

DOT code:

```
digraph {
0 [label=''Frontend Application'']; 1 [label=''Load Balancer'']; 2 [label=''Backend API Server'']; 3
    [label=''Authentication Module'']; 4 [label=''Business Logic Layer'']; 5 [label=''Database Connection
    Pool'']; 6 [label=''Primary Database'']; 7 [label=''Cache Layer'']; 8 [label=''Logging Service''];
0 → 1; 1 → 2; 2 → 3; 2 → 4; 3 → 5; 4 → 5; 5 → 6; 2 → 7; 2 → 8;
}
```

**Example 2**  Description: A microservices architecture where a load balancer distributes requests to multiple service instances, each connecting to its own database. The system includes service discovery, API gateway functionality, and inter-service communication through message queues.

Each microservice is containerized and deployed independently with its own database schema and business logic.

DOT code:

```
digraph {
0 [label=''API Gateway'']; 1 [label=''Load Balancer'']; 2 [label=''Service Discovery'']; 3 [label=''User
    Service'']; 4 [label=''Order Service'']; 5 [label=''Payment Service'']; 6 [label=''Notification
    Service'']; 7 [label=''User Database'']; 8 [label=''Order Database'']; 9 [label=''Payment Database''];
    10 [label=''Message Queue'']; 11 [label=''Cache Layer'']; 12 [label=''Monitoring Service'']; 13
    [label=''Config Service''];
0 → 1; 1 → 2; 2 → 3; 2 → 4; 2 → 5; 3 → 7; 4 → 8; 5 → 9; 3 → 10; 4 → 10; 5 → 10; 10 → 6; 3 → 11; 4
    → 11; 5 → 11; 2 → 12; 2 → 13;
}
```

**Example 3** Description: A data processing pipeline where data flows from multiple source systems through various processing stages including data validation, transformation, enrichment, and quality checks before being stored in a destination data warehouse. The pipeline includes error handling, monitoring, and retry mechanisms for fault tolerance.

DOT code:

```
digraph {
0 [label=''Source System A'']; 1 [label=''Source System B'']; 2 [label=''Source System C'']; 3
    [label=''Data Ingestion Layer'']; 4 [label=''Data Validation Module'']; 5 [label=''Data Transformation
    Engine'']; 6 [label=''Data Enrichment Service'']; 7 [label=''Quality Check Module'']; 8 [label=''Error
    Handler'']; 9 [label=''Retry Mechanism'']; 10 [label=''Data Warehouse'']; 11 [label=''Monitoring
    Dashboard'']; 12 [label=''Audit Log'']; 13 [label=''Metadata Store''];
0 → 3; 1 → 3; 2 → 3; 3 → 4; 4 → 5; 5 → 6; 6 → 7; 7 → 10; 4 → 8; 5 → 8; 6 → 8; 7 → 8; 8 → 9; 9 → 4; 3
    → 11; 7 → 12; 3 → 13;
}
```

**Example 4** Description: A message queue system where multiple producers send different types of messages to topic-based queues, and specialized consumers process these messages with dead letter handling, retry logic, and message persistence. The system includes consumer groups for load balancing and message ordering guarantees.

DOT code:

```
digraph {
0 [label=''Producer A'']; 1 [label=''Producer B'']; 2 [label=''Producer C'']; 3 [label=''Message Broker'']; 4
    [label=''Topic 1'']; 5 [label=''Topic 2'']; 6 [label=''Topic 3'']; 7 [label=''Consumer Group 1'']; 8
    [label=''Consumer Group 2'']; 9 [label=''Consumer A'']; 10 [label=''Consumer B'']; 11
    [label=''Consumer C'']; 12 [label=''Dead Letter Queue'']; 13 [label=''Retry Handler'']; 14
    [label=''Message Store'']; 15 [label=''Monitoring Service''];
0 → 3; 1 → 3; 2 → 3; 3 → 4; 3 → 5; 3 → 6; 4 → 7; 5 → 7; 6 → 8; 7 → 9; 7 → 10; 8 → 11; 9 → 12; 10 →
    12; 11 → 12; 12 → 13; 13 → 4; 3 → 14; 3 → 15;
}
```

**Example 5** Description: A comprehensive user authentication and authorization system where users login through multiple interfaces including web, mobile, and API endpoints. The system validates credentials against multiple identity providers, implements multi-factor authentication, generates and manages JWT tokens with refresh capabilities, maintains session state, and provides role-based access control with fine-grained permissions.

DOT code:

```
digraph {
0 [label=''Web Interface'']; 1 [label=''Mobile App'']; 2 [label=''API Gateway'']; 3 [label=''Authentication
    Service'']; 4 [label=''Identity Provider A'']; 5 [label=''Identity Provider B'']; 6 [label=''MFA
    Service'']; 7 [label=''SMS Provider'']; 8 [label=''Email Provider'']; 9 [label=''Token Generator'']; 10
```

```
      [label=''JWT Service'']; 11 [label=''Refresh Token Store'']; 12 [label=''Session Manager'']; 13
      [label=''Authorization Service'']; 14 [label=''Role Manager'']; 15 [label=''Permission Store'']; 16
      [label=''User Database'']; 17 [label=''Audit Logger'']; 18 [label=''Rate Limiter''];
0 → 3; 1 → 3; 2 → 3; 3 → 4; 3 → 5; 3 → 6; 6 → 7; 6 → 8; 3 → 9; 9 → 10; 10 → 11; 3 → 12; 3 → 13; 13
      → 14; 14 → 15; 3 → 16; 13 → 16; 3 → 17; 3 → 18;
}
```

## M    DETAILED DESCRIPTION OF MODEL ARCHITECTURES

Meta-Llama-3-8B-Instruct is part of Meta's Llama 3 family of open-weight large language models, trained on a mixture of publicly available and licensed data. The "Instruct" variant is fine-tuned with reinforcement learning from human feedback (RLHF) to follow instructions and generate helpful, safe, and coherent responses. It performs particularly well on reasoning, code generation, and instruction-following tasks, making it a strong choice for structured text-to-code generation scenarios.

Qwen2-7B-Instruct is an instruction-tuned version of Alibaba's Qwen2-7B model, trained on a diverse multilingual and multi-domain corpus. It is optimized for dialogue and instruction-following tasks, with strong performance on reasoning and code-related benchmarks. Its relatively small size and efficient architecture make it a good fit for tasks requiring both semantic understanding and structured output, especially in resource-constrained settings.

DeepSeek-LLM-7B-Chat is a 7B-parameter open-source model fine-tuned for conversational and code-related tasks. It is trained on a large-scale dataset with a focus on code understanding and generation, and excels at producing syntactically correct and semantically meaningful code snippets. Its chat-oriented fine-tuning also makes it robust to ambiguous or under-specified prompts, which is valuable in text-to-code generation where input descriptions can vary in clarity.

## N    PROMPTS AND RESULTS FOR LENGTHENED AND SHORTENED DESCRIPTIONS

To assess robustness, we generated lengthened and shortened paraphrased variants of the textual descriptions using GPT4o-1120. In the manual annotations set, the original descriptions contain ∼201 words, while lengthened and shortened paraphrased variants contain ∼599 and ∼139 words respectively. Similarly, in the test set, the original descriptions contain ∼203 words, while lengthened and shortened paraphrased variants contain ∼600 and ∼139 words respectively.

### N.1    GPT4O-1120 PROMPT FOR LENGTHENED DESCRIPTIONS

You will be given a short, cleaned description of an image. Your task is to significantly increase the length of the description **while** preserving ∗exactly∗ the same information content. Do not add any new details, facts, interpretations, architectural components, domain−specific terms, or any information not explicitly stated **in** the original description.

You must only:
− Rephrase, elaborate, and expand the phrasing of what is already present.
− Use more descriptive language, redundant clarification, or extended explanations.
− Maintain the original meaning strictly.
− Avoid introducing any task−related details, instructions, or references to datasets, DOT code, models, or any processes outside of the original text.

The output should be a verbose, highly expanded paraphrase that conveys the same information with no new content.

INPUT:
#CLEANED_DESCRIPTION_HERE#

OUTPUT:
A significantly longer version that preserves all original meaning and adds no new information.

## N.2 GPT4O-1120 Prompt for Shortened Descriptions

You will be given a cleaned description of an image. Your task is to shorten the description **while** preserving ∗all∗ of the information contained **in** the original text. You must not remove any factual content, omit steps, or lose meaning.

You must only:
– Compress, condense, and streamline the language.
– Merge sentences where possible without losing information.
– Remove redundant phrasing but keep all details present **in** the original.
– Avoid introducing any new content or changing meaning.
– Do not mention any task–specific details, DOT code, metadata, or anything outside the given description.

The output should be a shorter, more concise paraphrase that preserves every piece of original information.

INPUT:
*#CLEANED_DESCRIPTION_HERE#*

OUTPUT:
A shorter version that conveys all original content with no loss of meaning and no new additions.

## N.3 Results

The results are summarized in the Tables 5 and 6 for our best model (finetuned DeepSeek-7B) for the TEXT2ARCH manual annotation set and test set respectively. As expected, significant length shifts degrade performance, but the trends are consistent and provide valuable insights into the sensitivity of text-to-structure generation.

## O   Human Evaluation

For training Arch versus no-Arch classifier, we performed human labeling. Training data was filtered by annotators with deep domain knowledge (the authors), and the task is highly objective. For the manually annotated evaluation subset, the annotators labeled node and edge sets with very high consistency: 98 percent agreement for nodes and 96 percent for edges. Due to the cost of expert annotation, we did not perform overlapping annotation for the entire dataset, but the agreement numbers above reflect strong reliability.

We acknowledge the importance of human evaluation of our end to end system. Hence, we conducted a human preference study to assess diagram quality. Across paired comparisons, our model is preferred in 71.6% of the cases while DiagramAgent is preferred in 28.4%. This provides clear evidence that human evaluators perceive the outputs of our system as more semantically correct and structurally coherent.

## P   Additional Results

### P.1 Comparison with Automatikz

The results in Tables 7 and 8 show that the proposed Text2Arch model performs much better than Automatikz on both the manual annotation set as well as the test set.

### P.2 Compilation Success Rate

Compilation success is crucial for the practical utility of DOT-based generation systems. We computed compilation rates across all fine-tuned models. Interestingly, llama3 (40.03%) leads to much lower compilation rates compared to Qwen2 (95.67%) and DeepSeek models (93.44%). Our analysis shows that truncated output is the major problem.

### P.3 Varying Hungarian Algorithm Threshold ($\tau$)

String similarity matching may under- or over-match aliases. The purpose of the Hungarian matching with threshold $\tau = 0.5$ is limited to aligning lexical node labels during metric calculation.

In this section, we present full results across multiple thresholds ($\tau \in 0.1, 0.3, 0.5, 0.7, 0.9$), demonstrating monotonic behavior and showing that model rankings remain stable in Table 9.

## Q  Frequently Asked Questions

### Q.1  Definition of "scientific architecture"

We use the term "scientific architecture diagrams" to refer to structured graphical representations of computational or experimental workflows, methodological pipelines, and scientific system designs commonly found in scientific and engineering publications.

### Q.2  For the same diagram, only one correct DOT exists?

A single diagram can indeed correspond to multiple syntactically different DOT files due to variations in node ordering, edge ordering, and layout directives. To provide a unique training target and ensure valid comparisons, we canonicalize all DOT graphs prior to use. Specifically, we: (i) sort nodes lexicographically by label, (ii) sort edges by source–target index pairs, (iii) remove layout or stylistic attributes that do not affect topology.

Following the canonicalization procedure, each diagram is mapped to a single DOT representation that uniquely captures its structure. This ensures consistent supervision and unambiguous evaluation, while remaining correct up to graph isomorphism. The evaluation metrics are therefore insensitive to semantically irrelevant syntactic variations.

During evaluation, we compute structural equivalence by converting DOT graphs into adjacency matrices and performing graph isomorphism checks. Two diagrams are considered equivalent if their node labels and directed edges correspond under a bijection. All reported graph-level metrics operate strictly on node content and edge connectivity and are invariant to layout differences.

This approach is consistent with evaluation strategies used in other structured-generation domains and additionally, our text based evaluations align with the evaluations adopted by DiagramAgent (Wei et al., 2024), which we use as a baseline.

### Q.3  Why was GPT5 not used?

GPT-5 was released just before the ICLR submission deadline. GPT-5.1 was also released post-deadline. For fairness and compliance, we included models readily available before the submission deadline.

### Q.4  Did you de-duplicate near-identical figures/descriptions across splits?

We designed the dataset split to be random, ensuring no intentional overlap. Also, note that our dataset contains data from real papers hosted on arxiv. Unlike web collections where there could be duplicates, typically it is not expected that figures would repeat across different papers on arxiv.

## R  Limitations

While our work makes significant strides, it has several limitations. First, our dataset and experiments are limited to English-language descriptions, and extending to multilingual settings remains unexplored. Second, we did not experiment with advanced alignment techniques such as Direct Preference Optimization (DPO) or Reinforcement Learning with Human Feedback (RLHF), which could further improve human-perceived quality. Third, our models are trained and evaluated on clean, well-aligned data; their robustness to noisy or ambiguous real-world inputs is yet to be assessed. Fourth,

while we focused on DOT code as the intermediate representation, exploring alternative graph description languages or direct image generation remains an open direction.

Additionally, in some complex cases, our trained models are unable to generate architecture diagrams with dense connections, multiple self-loops. Further, the DOT code generation pipeline relies heavily on GPT-4o and rule-based post-processing, which may introduce inconsistencies or propagate errors from OCR and object detection stages, especially in complex or low-quality diagrams. DOT2 edge direction relies on arrowhead heuristics; failure modes although not commonly observed (overlapping arrows, ambiguous heads, dashed lines) can occur in some cases. While we categorize diagrams into easy, medium, and hard based on node count, this does not always reflect true semantic or structural complexity.

Multi-edges and duplicate edges are not explicitly handled or deduplicated.

Addressing these limitations can further enhance the applicability and generalization of text-to-architecture systems.

## S    ETHICS CONSIDERATIONS AND ASSOCIATED RISKS

While our work advances the state of text-to-architecture diagram generation, it is important to acknowledge potential ethical implications and risks. First, our dataset and models are trained exclusively on English-language data, which may limit accessibility and fairness for non-English speakers and perpetuate linguistic biases. Second, the models may inadvertently encode and amplify biases present in the training data, such as over-representing certain architectural patterns or terminology, which could lead to homogenized or culturally skewed outputs. Third, the generated diagrams, while structurally faithful, are not guaranteed to be semantically correct in all contexts, and over-reliance on automated outputs without human verification could result in misleading or erroneous designs. Additionally, the models are not explicitly aligned with human preferences through techniques like DPO or RLHF, which may limit their alignment with user intent in ambiguous cases. Finally, as with any generative system, there is a risk of misuse, such as generating plausible-looking but incorrect diagrams for critical systems, which could have downstream safety or security implications. We recommend that users treat the outputs as assistive rather than authoritative and incorporate human oversight in high-stakes scenarios.

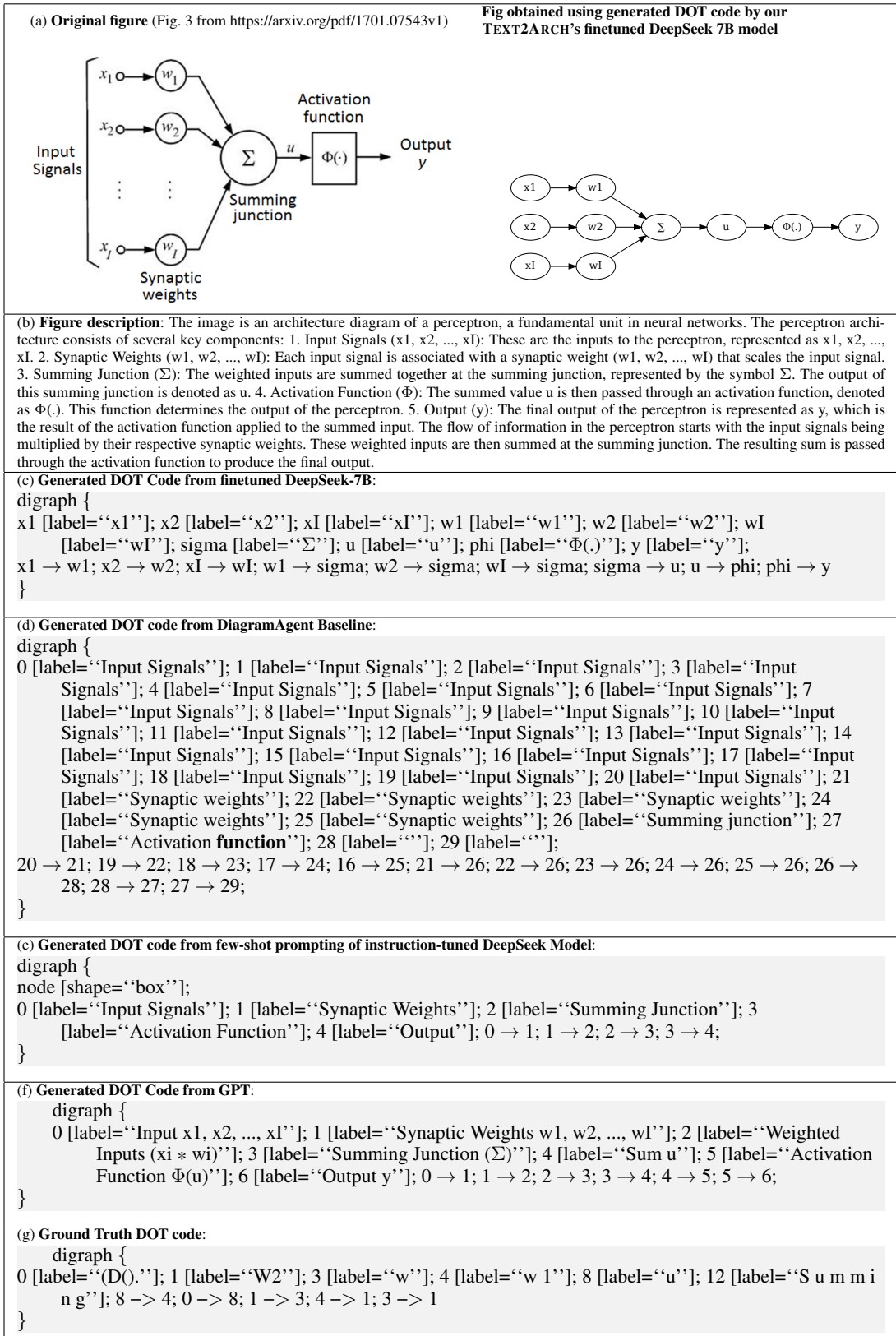

(a) **Original figure** (Fig. 3 from https://arxiv.org/pdf/1701.07543v1)

**Fig obtained using generated DOT code by our TEXT2ARCH's finetuned DeepSeek 7B model**

(b) **Figure description**: The image is an architecture diagram of a perceptron, a fundamental unit in neural networks. The perceptron architecture consists of several key components: 1. Input Signals (x1, x2, ..., xI): These are the inputs to the perceptron, represented as x1, x2, ..., xI. 2. Synaptic Weights (w1, w2, ..., wI): Each input signal is associated with a synaptic weight (w1, w2, ..., wI) that scales the input signal. 3. Summing Junction ($\Sigma$): The weighted inputs are summed together at the summing junction, represented by the symbol $\Sigma$. The output of this summing junction is denoted as u. 4. Activation Function ($\Phi$): The summed value u is then passed through an activation function, denoted as $\Phi(.)$. This function determines the output of the perceptron. 5. Output (y): The final output of the perceptron is represented as y, which is the result of the activation function applied to the summed input. The flow of information in the perceptron starts with the input signals being multiplied by their respective synaptic weights. These weighted inputs are then summed at the summing junction. The resulting sum is passed through the activation function to produce the final output.

(c) **Generated DOT Code from finetuned DeepSeek-7B**:

```
digraph {
x1 [label=``x1''']; x2 [label=``x2'']; xI [label=``xI'']; w1 [label=``w1'']; w2 [label=``w2'']; wI
    [label=``wI'']; sigma [label=``Σ'']; u [label=``u'']; phi [label=``Φ(.)'']; y [label=``y''];
x1 → w1; x2 → w2; xI → wI; w1 → sigma; w2 → sigma; wI → sigma; sigma → u; u → phi; phi → y
}
```

(d) **Generated DOT code from DiagramAgent Baseline**:

```
digraph {
0 [label=``Input Signals'']; 1 [label=``Input Signals'']; 2 [label=``Input Signals'']; 3 [label=``Input
    Signals'']; 4 [label=``Input Signals'']; 5 [label=``Input Signals'']; 6 [label=``Input Signals'']; 7
    [label=``Input Signals'']; 8 [label=``Input Signals'']; 9 [label=``Input Signals'']; 10 [label=``Input
    Signals'']; 11 [label=``Input Signals'']; 12 [label=``Input Signals'']; 13 [label=``Input Signals'']; 14
    [label=``Input Signals'']; 15 [label=``Input Signals'']; 16 [label=``Input Signals'']; 17 [label=``Input
    Signals'']; 18 [label=``Input Signals'']; 19 [label=``Input Signals'']; 20 [label=``Input Signals'']; 21
    [label=``Synaptic weights'']; 22 [label=``Synaptic weights'']; 23 [label=``Synaptic weights'']; 24
    [label=``Synaptic weights'']; 25 [label=``Synaptic weights'']; 26 [label=``Summing junction'']; 27
    [label=``Activation function'']; 28 [label=``'']; 29 [label=``''];
20 → 21; 19 → 22; 18 → 23; 17 → 24; 16 → 25; 21 → 26; 22 → 26; 23 → 26; 24 → 26; 25 → 26; 26 →
    28; 28 → 27; 27 → 29;
}
```

(e) **Generated DOT code from few-shot prompting of instruction-tuned DeepSeek Model**:

```
digraph {
node [shape=``box''];
0 [label=``Input Signals'']; 1 [label=``Synaptic Weights'']; 2 [label=``Summing Junction'']; 3
    [label=``Activation Function'']; 4 [label=``Output'']; 0 → 1; 1 → 2; 2 → 3; 3 → 4;
}
```

(f) **Generated DOT Code from GPT**:

```
    digraph {
    0 [label=``Input x1, x2, ..., xI'']; 1 [label=``Synaptic Weights w1, w2, ..., wI'']; 2 [label=``Weighted
        Inputs (xi * wi)'']; 3 [label=``Summing Junction (Σ)'']; 4 [label=``Sum u'']; 5 [label=``Activation
        Function Φ(u)'']; 6 [label=``Output y'']; 0 → 1; 1 → 2; 2 → 3; 3 → 4; 4 → 5; 5 → 6;
}
```

(g) **Ground Truth DOT code**:

```
    digraph {
0 [label=``(D().'']; 1 [label=``W2'']; 3 [label=``w'']; 4 [label=``w 1'']; 8 [label=``u'']; 12 [label=``S u m m i
    n g'']; 8 −> 4; 0 −> 8; 1 −> 3; 4 −> 1; 3 −> 1
}
```

Figure 4: Case Study 1: Comparison showing DeepSeek-7B inference significantly outperforming all baseline methods. DeepSeek achieved superior performance across all key metrics (node_f1, edge_f1, node_accuracy) compared to diagramAgent, GPT, and DeepSeek-chat baselines.

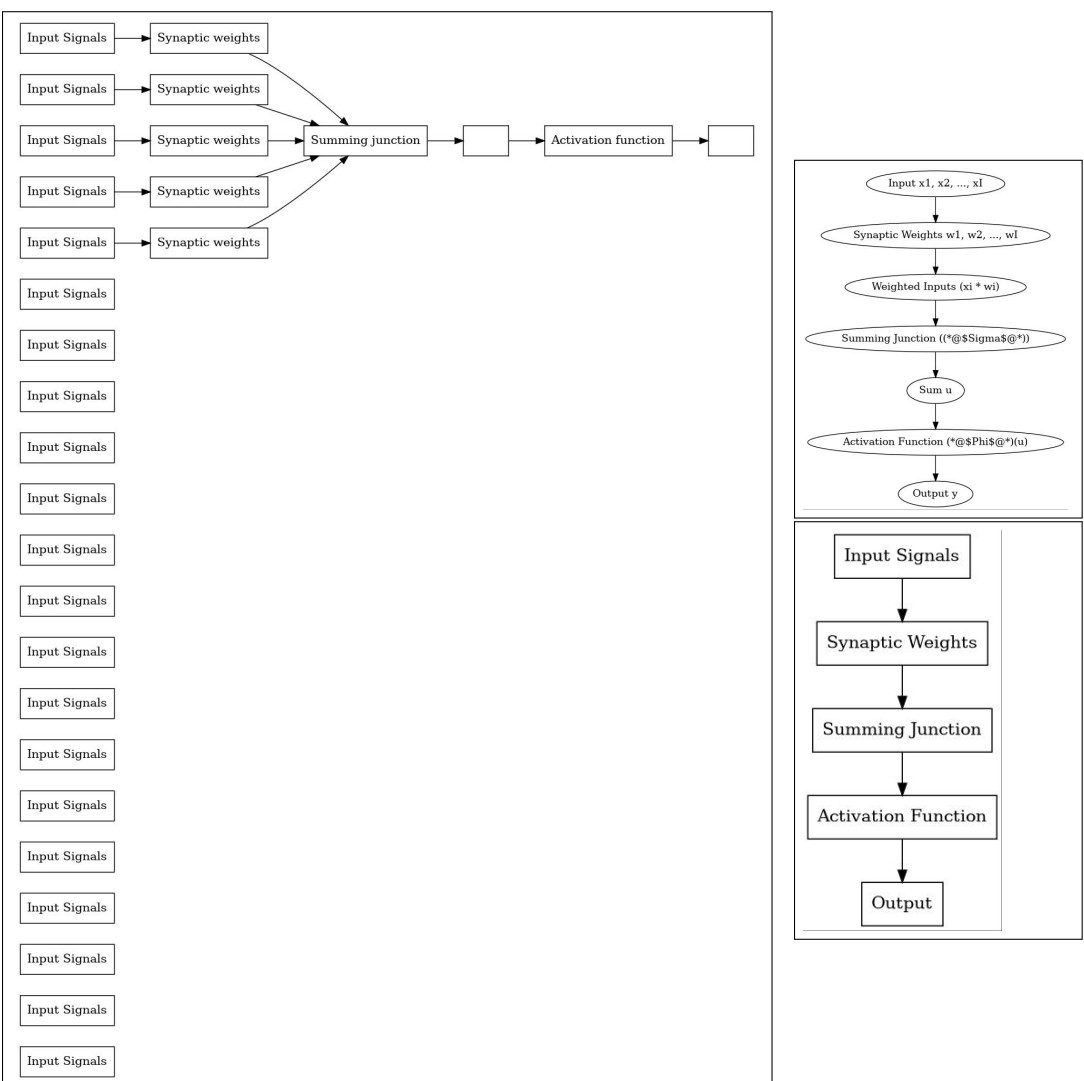

Figure 5: Illustrations for generated dots using DiagramAgent (left), GPT (right top) and fewShot DeepSeek (right bottom) corresponding to Case Study 1 shown in Fig. 4.

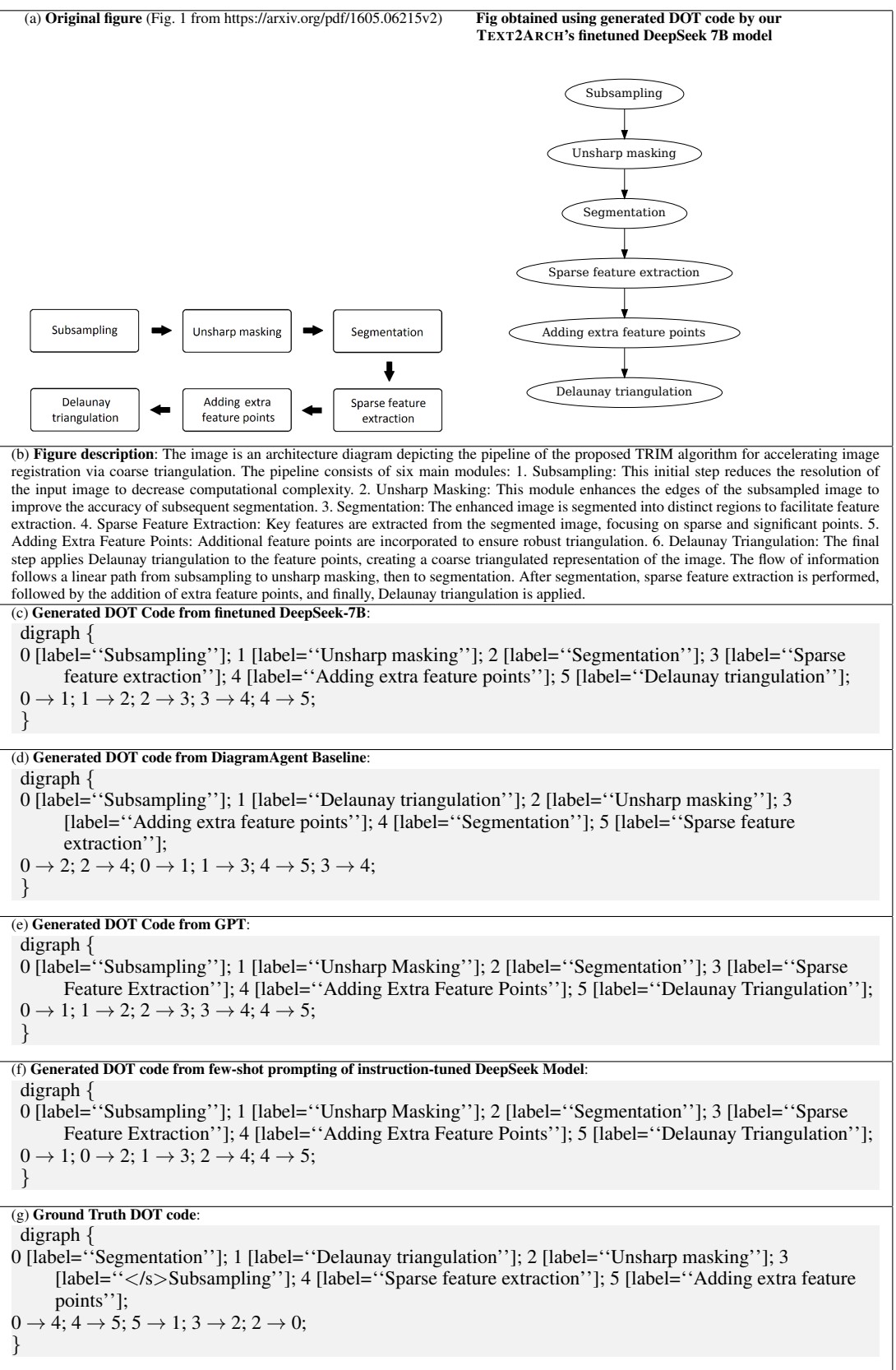

Figure 6: Case Study 2: Comparison showing DeepSeek-7B inference significantly outperforming all baseline methods. DeepSeek successfully generated DOT code that exactly matches the ground truth structure while other baselines show varying degrees of performance.

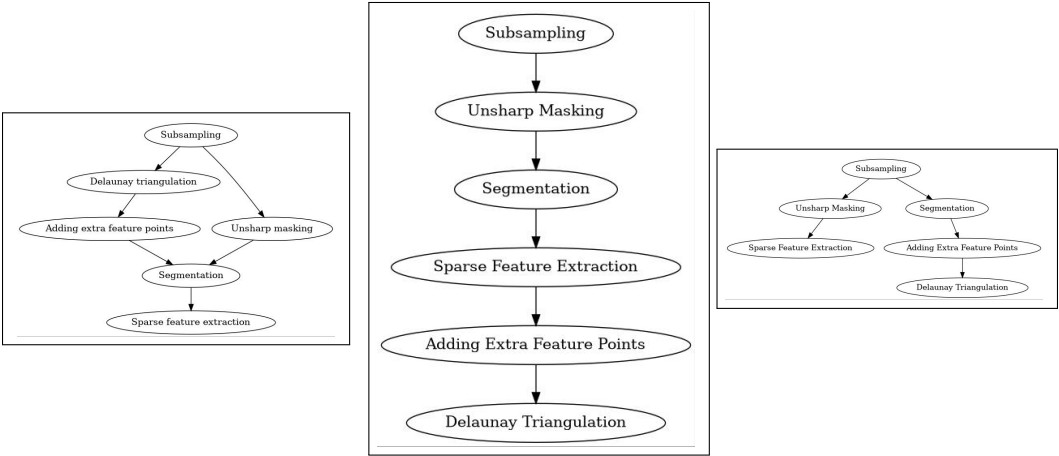

Figure 7: Illustrations for generated dots using DiagramAgent (left), GPT (middle) and fewShot DeepSeek (right) corresponding to Case Study 2 shown in Fig. 6.

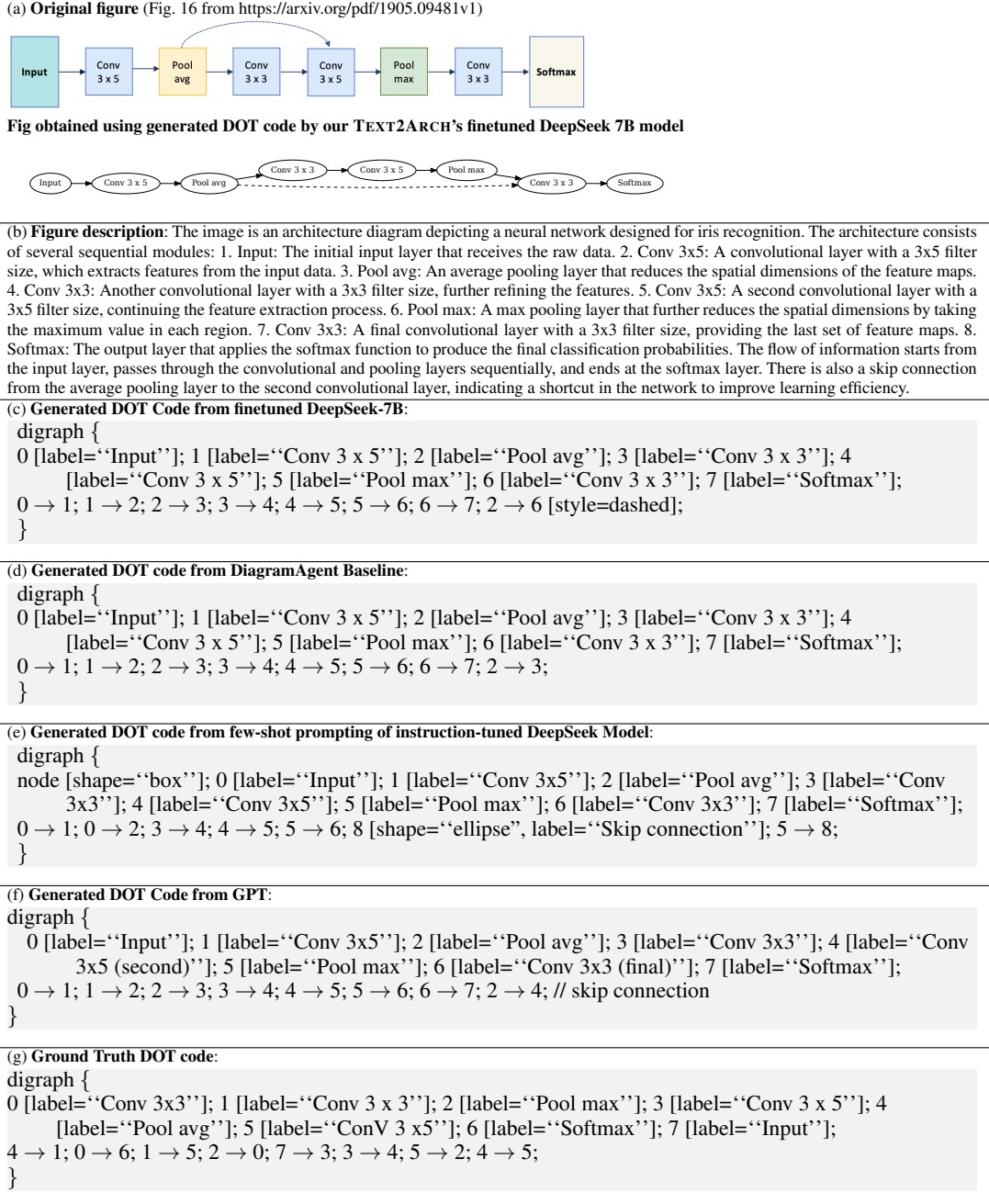

Figure 8: Case Study 3: Comparison showing DeepSeek-7B inference significantly outperforming all baseline methods. successfully generated DOT code that exactly matches the ground truth structure while other baselines show varying degrees of performance.

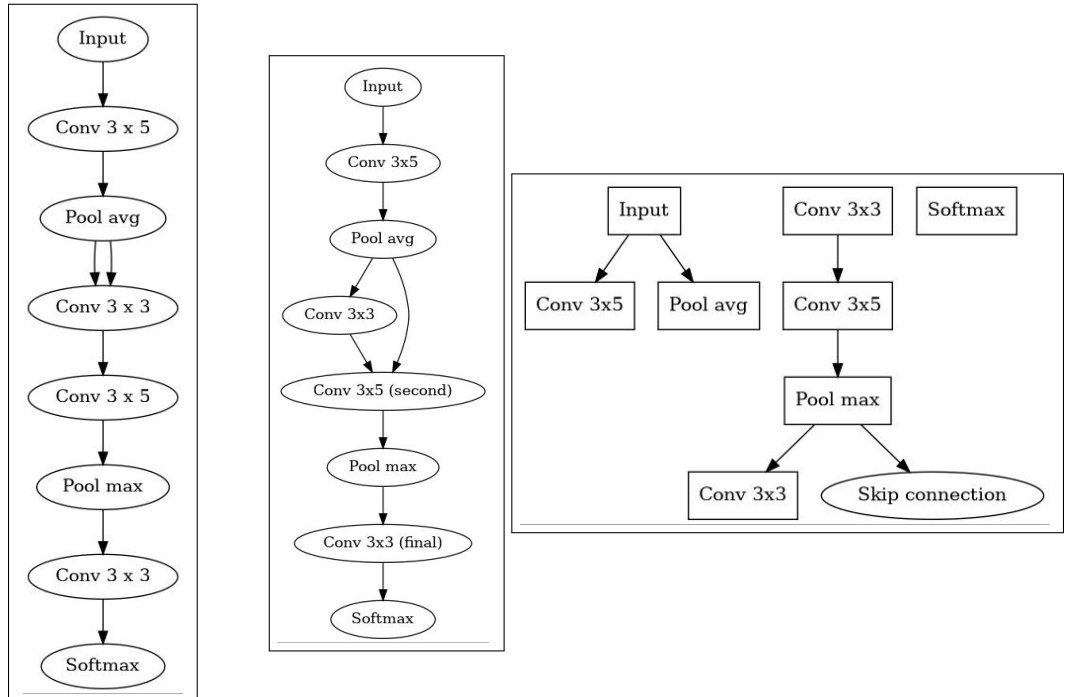

Figure 9: Illustrations for generated dots using DiagramAgent (left), GPT (middle) and fewShot DeepSeek (right) corresponding to Case Study 3 shown in Fig. 8.

| | Text Metrics | | | | Graph Metrics | | | | | | | | |
|---|---|---|---|---|---|---|---|---|---|---|---|---|---|
| | ROUGE-L | CodeBLEU | Edit Distance | chrF | Node Prec | Node Recall | Node F1 | Node PR-AUC | Edge Prec | Edge Recall | Edge F1 | Edge PR-AUC | Jaccard Sim. |
| Lengthened Desc | 44.7 | 37.2 | 682 | 48.7 | 72.8 | 62.7 | 64.4 | 23.8 | 52.3 | 35.0 | 40.2 | 25.1 | 31.2 |
| Shortened Desc | 55.3 | 50.0 | 491 | 65.0 | 70.6 | 75.5 | 71.4 | 19.8 | 61.9 | 45.3 | 50.7 | 34.0 | 39.6 |
| Original Desc. | 55.2 | 49.3 | 407 | 66.6 | 66.1 | 78.1 | 69.4 | 27.4 | 59.4 | 44.6 | 49.1 | 35.1 | 39.8 |

Table 5: Description length variation results on TEXT2ARCH manual annotation set.

| | Text Metrics | | | | Graph Metrics | | | | | | | | |
|---|---|---|---|---|---|---|---|---|---|---|---|---|---|
| | ROUGE-L | CodeBLEU | Edit Distance | chrF | Node Prec | Node Recall | Node F1 | Node PR-AUC | Edge Prec | Edge Recall | Edge F1 | Edge PR-AUC | Jaccard Sim. |
| Lengthened Desc | 32.2 | 28.3 | 795 | 32.3 | 30.6 | 25.4 | 26.3 | 10.4 | 16.9 | 9.2 | 11.1 | 10.0 | 8.0 |
| Shortened Desc | 33.7 | 32.5 | 723 | 40.5 | 25.2 | 28.4 | 25.3 | 8.4 | 16.5 | 9.6 | 11.3 | 7.9 | 8.2 |
| Original Desc. | 46.8 | 34.5 | 608 | 55.7 | 66.2 | 69.6 | 65.7 | 21.5 | 46.4 | 34.2 | 38.0 | 23.7 | 28.6 |

Table 6: Description length variation results on TEXT2ARCH test set.

| | Text Metrics | | | | Graph Metrics | | | | | | | | |
|---|---|---|---|---|---|---|---|---|---|---|---|---|---|
| | ROUGE-L | Edit Distance | chrF | Node Prec | Node Recall | Node F1 | Node PR-AUC | Edge Prec | Edge Recall | Edge F1 | Edge PR-AUC | Jaccard Sim. |
| Automatikz | 41.5 | 531 | 41.2 | 46.9 | 36.9 | 38.7 | 12.3 | 19.1 | 8.6 | 10.9 | 8.9 | 7.2 |
| Text2Arch (fine-tuned DeepSeek-7B) | 55.2 | 407 | 66.6 | 66.1 | 78.1 | 69.4 | 27.4 | 59.4 | 44.6 | 49.1 | 35.1 | 39.8 |

Table 7: Results on TEXT2ARCH manual annotation set.

| | Text Metrics | | | | Graph Metrics | | | | | | | | |
|---|---|---|---|---|---|---|---|---|---|---|---|---|---|
| | ROUGE-L | Edit Distance | chrF | Node Prec | Node Recall | Node F1 | Node PR-AUC | Edge Prec | Edge Recall | Edge F1 | Edge PR-AUC | Jaccard Sim. |
| Automatikz | 37.9 | 608 | 37.0 | 42.1 | 33.6 | 34.5 | 11.3 | 20.4 | 8.4 | 10.8 | 8.3 | 7.2 |
| Text2Arch (fine-tuned DeepSeek-7B) | 46.8 | 608 | 55.7 | 66.2 | 69.6 | 65.7 | 21.5 | 46.4 | 34.2 | 38 | 23.7 | 28.6 |

Table 8: Results on TEXT2ARCH test set.

| Model | Node Prec | Node Recall | Node F1 | Node PR-AUC | Edge Prec | Edge Recall | Edge F1 | Edge PR-AUC | Jaccard Sim |
|---|---|---|---|---|---|---|---|---|---|
| Llama-3-8B ($\tau$=0.1) | 31.2 | 49.1 | 35.2 | 6.9 | 23.2 | 15 | 17.5 | 8.7 | 11.3 |
| Llama-3-8B ($\tau$=0.3) | 28 | 44.2 | 31.6 | 6.9 | 21.9 | 9.6 | 12.6 | 8.7 | 7.9 |
| Llama-3-8B ($\tau$=0.5) | 22.7 | 35.8 | 25.5 | 6.9 | 20.2 | 6.1 | 8.7 | 8.7 | 5.5 |
| Llama-3-8B ($\tau$=0.7) | 19.1 | 30.4 | 21.5 | 6.9 | 17.6 | 4.6 | 6.7 | 8.7 | 4.2 |
| Llama-3-8B ($\tau$=0.9) | 10.2 | 15.7 | 11.3 | 6.9 | 8.4 | 1.7 | 2.7 | 8.7 | 1.7 |
| Qwen2-7B ($\tau$=0.1) | 35.8 | 52.9 | 39.2 | 7.9 | 20.4 | 13.4 | 15.5 | 8.1 | 9.9 |
| Qwen2-7B ($\tau$=0.3) | 33 | 48.7 | 36.1 | 7.9 | 20.7 | 10.1 | 12.8 | 8.1 | 8.2 |
| Qwen2-7B ($\tau$=0.5) | 28.4 | 41.7 | 31 | 7.9 | 21.4 | 7.5 | 10.4 | 8.1 | 6.6 |
| Qwen2-7B ($\tau$=0.7) | 25.1 | 36.8 | 27.4 | 7.9 | 20.2 | 6.2 | 8.7 | 8.1 | 5.6 |
| Qwen2-7B ($\tau$=0.9) | 15.1 | 21.6 | 16.4 | 7.9 | 13.1 | 3.3 | 4.8 | 8.1 | 3 |
| DeepSeek-7B ($\tau$=0.1) | 67.3 | 69.9 | 66.3 | 20.8 | 42.6 | 34.3 | 37 | 22.5 | 27.6 |
| DeepSeek-7B ($\tau$=0.3) | 65.7 | 68.2 | 64.7 | 20.8 | 44.7 | 32.6 | 36.3 | 22.5 | 27.1 |
| DeepSeek-7B ($\tau$=0.5) | 62.8 | 65.2 | 61.8 | 20.8 | 47.4 | 30.6 | 35.3 | 22.5 | 26.3 |
| DeepSeek-7B ($\tau$=0.7) | 60 | 62.3 | 59 | 20.8 | 47.4 | 28.3 | 33.3 | 22.5 | 24.7 |
| DeepSeek-7B ($\tau$=0.9) | 43.6 | 44.9 | 42.9 | 20.8 | 39.8 | 18.5 | 22.9 | 22.5 | 16.7 |

Table 9: Graph Metrics with Varying Hungarian Algorithm Threshold $\tau$

