# OpenReview forum: "Text2Arch: A Dataset for Generating Scientific Architecture Diagrams from Natural Language Descriptions"
_ICLR.cc/2026/Conference — ICLR 2026 Poster_

### Official Review · Reviewer_yJFJ · 2025-10-27

**Soundness:** 3
**Presentation:** 3
**Contribution:** 3
**Rating:** 6
**Confidence:** 4

**Summary:**

The paper introduces TEXT2ARCH, a new task and large-scale dataset (75,127 text–DOT–image triplets; 60,519/7,565/7,043 split) for generating scientific architecture diagrams from natural-language descriptions via intermediate DOT code. The dataset is curated through a three-stage pipeline—(1) filtering architecture figures, (2) DOT extraction with OCR/object detection + GPT refinement (DOT1→DOT2→DOT3), and (3) description refinement—illustrated in Fig. 2, with distributions and statistics in Fig. 3. The authors propose graph-level metrics (node/edge precision/recall/F1, PR-AUC, Jaccard) in addition to NLG metrics.

**Strengths:**

- Scoped, high-impact dataset for architecture diagrams with aligned text–code–image triplets and complexity bucketing; useful beyond the paper’s models.
- Clear curation pipeline (classifier + OCR/detection + GPT refinement) with ablations across DOT variants (DOT3≫DOT1/DOT2).
- Consistent empirical gains from finetuning small models (DeepSeek‑7B best on both automatic and GPT-based judging).

**Weaknesses:**

- Baseline fairness: DiagramAgent (TikZ) → DOT via GPT may degrade/alter structure; results could change with a native TikZ-based evaluation.
- Label/eval circularity risk: GPT‑4o is used to generate/refine DOT labels (DOT1→DOT3) and to score outputs; this can bias comparisons and makes “ground truth” partly model-dependent. The human set (n=99) helps but is small.
- string-similarity matching (Hungarian with τ=0.5) ignores diagram layout and may over/under-match aliases; multi-edges and duplicates aren’t handled; layout attributes are ignored.
- no analysis of near-duplicate figures across splits or of overlap between description sources and DOT generation that could inflate performance.

**Questions:**

- Beyond Table 3, can you expand the human-labeled set (≥500) and report the same metrics to better calibrate DOT3 quality?
- Provide TikZ-native evaluation for DiagramAgent (and a DOT-native variant for your model) to avoid cross-format conversion via GPT.
- Did you de-duplicate near-identical figures/descriptions across splits? Please report a hash/similarity analysis.
- Show results across node-match thresholds and with alternative label normalization; report effects of handling multi-edges/duplicates.
- Quantify contributions of each curation stage (classifier, OCR/detection, GPT refinements) to final performance; show training with DOT1-only/DOT2-only.

---

> ### Author Response · Authors · 2025-12-02
>
> We thank the reviewer for their time and effort. Also, we are grateful for the thought-provoking comments.
>
> > W1: Baseline fairness: TikZ → DOT conversion for DiagramAgent
>
> We agree that DiagramAgent natively outputs TikZ, and converting its TikZ code to DOT using GPT may introduce structural drift. To more fairly compare methods across their native representation spaces, we evaluated the Diagram Agent model using Tikz code itself. We used GPT4o to convert ground truth dot files into Tikz domain, and then computed the evaluation metrics for diagram agent in Tikz domain for both manual labelled set as well as ground truth.
>
> The performance was very poor (even after standardized conversion) confirming that TikZ pipelines struggle with semantic reconstruction:
>
> * Test set: Node F1 = 23.1, Edge F1 = 7.2, ROUGE-L = 15.4, CodeBLEU = 11.0
> * Manual annotation set: Node F1 = 22.1, Edge F1 = 13.3, ROUGE-L = 6.1, CodeBLEU = 12.3
>
> These results are significantly below both GPT-4o and our finetuned Text2Arch models.
>
> A full TikZ evaluation would require constructing a text-to-TikZ dataset or designing a robust DOT↔TikZ conversion pipeline, which is beyond the scope of the present work, but we appreciate the reviewer’s suggestion and highlight it as an interesting future direction.
>
> Also, as noted in the paper: DOT emphasizes semantic structure, node–edge relationships, and graph correctness. TikZ emphasizes visual styling and rendering control. For scientific architecture diagrams, workflow systems, computational pipelines, and structured process diagrams, semantic correctness is more important than stylistic layout. DOT is the natural representation for such tasks.
>
> > W2: Label/evaluation circularity: GPT-4o in labeling and evaluation
>
> We appreciate the reviewer’s concern regarding potential circularity. However, the ground truth is not directly GPT-generated.
> * DOT1 is partially GPT-derived, but as shown in Table 3, DOT1 is not used as ground truth.
> * DOT2 is produced via (a) Florence OCR extraction and (b) Flowchart Object Detection (Shukla et al., 2025), which yields a structured set of nodes and edges independent of GPT.
> * DOT3 refines DOT2 using GPT-4o, but GPT operates only as a consistency filter not as a standalone generator.
>
> Thus, ground truth = DOT3 = (Detector outputs + GPT refinement), not GPT-only.
>
> Further, human-labeled set demonstrates independence. The manual annotation set of 99 diagrams consists of 1127 nodes and 1170 edges labeled entirely by domain-expert annotators. This provides a fully GPT-free benchmark subset for evaluating potential bias.
>
> > W3: Matching procedure and Hungarian algorithm threshold ($\tau$)
>
> We agree that string similarity matching may under- or over-match aliases, and thank the reviewer for pointing this out. The purpose of the Hungarian matching with threshold τ = 0.5 is limited to aligning lexical node labels during metric calculation; layout fidelity and rendering differences are intentionally ignored because the task concerns semantic structure, not spatial layout.
>
> To address the concern, we have now included full results across multiple thresholds (τ ∈ {0.1, 0.3, 0.5, 0.7, 0.9}), demonstrating monotonic behavior and showing that model rankings remain stable. We use standard normalization steps like lowercasing and whitespace removal. We have included this now in Appendix R.3 in the revised submission.
>
> “Multi-edges and duplicate edges are not explicitly handled or deduplicated.” – We already mentioned this in the Appendix section on Hyper-parameters in the original submission. Now we have included this in the limitations section also. Layout attributes (rank, pos, etc.) are intentionally ignored to focus on structural correctness.
>
> ||Node Prec|Node Recall|Node F1|Node PR-AUC|Edge Prec|Edge Recall|Edge F1|Edge PR-AUC|Jaccard Sim|
> |---------------------|--------|--------|--------|--------|--------|--------|--------|--------|--------|
> |Llama-3-8B ($\tau$=0.1)|31.2|49.1|35.2|6.9|23.2|15|17.5|8.7|11.3|
> |Llama-3-8B ($\tau$=0.3)|28|44.2|31.6|6.9|21.9|9.6|12.6|8.7|7.9|
> |Llama-3-8B ($\tau$=0.5)|22.7|35.8|25.5|6.9|20.2|6.1|8.7|8.7|5.5|
> |Llama-3-8B ($\tau$=0.7)|19.1|30.4|21.5|6.9|17.6|4.6|6.7|8.7|4.2|
> |Llama-3-8B ($\tau$=0.9)|10.2|15.7|11.3|6.9|8.4|1.7|2.7|8.7|1.7|
> |Qwen2-7B ($\tau$=0.1)|35.8|52.9|39.2|7.9|20.4|13.4|15.5|8.1|9.9|
> |Qwen2-7B ($\tau$=0.3)|33|48.7|36.1|7.9|20.7|10.1|12.8|8.1|8.2|
> |Qwen2-7B ($\tau$=0.5)|28.4|41.7|31|7.9|21.4|7.5|10.4|8.1|6.6|
> |Qwen2-7B ($\tau$=0.7)|25.1|36.8|27.4|7.9|20.2|6.2|8.7|8.1|5.6|
> |Qwen2-7B ($\tau$=0.9)|15.1|21.6|16.4|7.9|13.1|3.3|4.8|8.1|3|
> |DeepSeek-7b ($\tau$=0.1)|67.3|69.9|66.3|20.8|42.6|34.3|37|22.5|27.6|
> |DeepSeek-7b ($\tau$=0.3)|65.7|68.2|64.7|20.8|44.7|32.6|36.3|22.5|27.1|
> |DeepSeek-7b ($\tau$=0.5)|62.8|65.2|61.8|20.8|47.4|30.6|35.3|22.5|26.3|
> |DeepSeek-7b ($\tau$=0.7)|60|62.3|59|20.8|47.4|28.3|33.3|22.5|24.7|
> |DeepSeek-7b ($\tau$=0.9)|43.6|44.9|42.9|20.8|39.8|18.5|22.9|22.5|16.7|

---

> > ### Author Response · Authors · 2025-12-02
> >
> > > W4: Near-duplicate figures across splits
> >
> > We designed the dataset split to be random, ensuring no intentional overlap. Also, note that our dataset contains data from real papers hosted on arxiv. Unlike web collections where there could be duplicates, typically it is not expected that figures would repeat across different papers on arxiv.
> >
> > > Q1: Expanding the human-labeled set beyond 99 examples
> >
> > We fully agree that larger human-labeled sets provide stronger calibration. However, each annotated diagram contains on average 11.16 nodes and 11.58 edges.
> >
> > Annotators must identify node text, node types, all directed edges, nested structures, block boundaries. Easy samples requires ≈15 minutes to label; hard samples take up to one hour.
> >
> > A 500-example set would require >300 annotator-hours. Given the high cost of expert annotation, we selected a stratified set (easy/medium/hard), which already yields more than 1100 node labels and 1170 edge labels.
> >
> > > Q2: TikZ-native evaluation and DOT-native evaluation
> >
> > Same as W1. Answered above.
> >
> > > Q3: de-duplicate near-identical figures/descriptions across splits
> >
> > Same as W4. Answered above.
> >
> > > Q4: Node-match thresholds, normalization, multi-edge handling
> >
> > Same as W3. Answered above.
> >
> > > Q5: Quantifying contributions of each curation stage (DOT1, DOT2, DOT3)
> >
> > Table 3 already shows that DOT3 >> DOT2 >> DOT1
> >
> > We agree that training on DOT1 or DOT2 would be ideal, but finetuning requires >5 days on 8 V100 GPUs per run. Conducting three full training runs per model (DOT1/DOT2/DOT3) across all architectures is computationally infeasible given compute scarcity on our side.
> >
> > Our strategy was therefore:
> > * Evaluate DOT1, DOT2, DOT3 using GPT-4o and human-labeled subsets.
> > * Identify DOT3 as the best-quality representation.
> > * Train models only on DOT3 and show results in Tables 1 and 2.
> >
> > We are confident about including detailed finetuning results with DOT1 and DOT2 in the camera ready version.

---

### Official Review · Reviewer_uvQ1 · 2025-10-28

**Soundness:** 2
**Presentation:** 2
**Contribution:** 3
**Rating:** 6
**Confidence:** 3

**Summary:**

**TEXT2ARCH** introduces a 75K+ text–DOT–image dataset that fills a clear gap for text-to-architecture generation, compares **DOT1/2/3** variants, and adds **novel graph-level metrics**; fine-tuned 7B–8B models (notably DeepSeek-7B) beat DiagramAgent and few-shot baselines.

**Strengths:**

1. **TEXT2ARCH fills a clear data gap.** The dataset addresses a previously missing resource for text-to-architecture diagram generation with aligned text–code–image triplets.
2. **Comprehensive comparison of DOT variants.** Evaluating DOT1/DOT2/DOT3 provides a thorough view of how different curation/refinement stages affect quality.
3. **Novel, task-appropriate metrics.** The graph-level evaluation (node/edge F1, PR-AUC, Jaccard) is thoughtful and well-aligned with the problem’s structural nature.

**Weaknesses:**

1. **Compilation success rate is unreported.** The paper does not quantify the percentage of generated DOT that compiles successfully, which is critical for practical usability.
2. **Limited qualitative evidence.** Case studies mostly show prompts and code; they lack rich visual side-by-side outputs and analyses of both good and bad generations across compared methods. It would be stronger to show the rendered diagrams and juxtapose multiple model outputs with brief error analyses.
3. **Insufficient SFT details (around line 322).** SFT setup is under-specified, only the prompt is given. Training loss/objectives should be described to ensure reproducibility.
4. **Missing TikZ results despite related discussion.** While focusing on DOT is reasonable, the paper cites TikZ-based prior work; a small TikZ transfer study (even limited) would help position the approach and set expectations for broader applicability.

**Questions:**

See weaknesses; I’m open to revising the score if the rebuttal provides sufficient evidence.

---

> ### Author Response · Authors · 2025-12-02
>
> We thank the reviewer for the constructive feedback and for noting several strengths, including the dataset contribution, the comparative analysis of DOT variants, and the introduction of structured graph-level metrics. We address each concern in detail below and incorporate the clarifications and additional results in the revised manuscript.
>
> > W1: Compilation success rate
>
> We agree that compilation success is crucial for the practical utility of DOT-based generation systems. We computed compilation rates across all fine-tuned models. Interestingly, llama3 (40.03%) leads to much lower compilation rates compared to Qwen2 (95.67%) and DeepSeek models (93.44%). Our analysis shows that truncated output is the major problem. We have included this now in Appendix R.2 in the revised submission.
>
> > W2: Limited qualitative evidence
>
> We thank the reviewer for this suggestion. Our original submission focused on code-level and metric-based analysis, but we agree that richer qualitative results will help readers better understand model behavior. In the initial paper pdf, we had included 3 case studies which included illustrations for the original figure and graph generated using our finetuned DeepSeek 7B model. In the revised version, we have now included side-by-side visual diagrams generated by all evaluated models (DiagramAgent, GPT, and our few-shot prompting based DeepSeek model). See Figures 5,7 and 9 in the revised submission. The improvements obtained using our model are very clear from these illustrations. Baseline methods lead to several errors like structural mismatches, missing nodes, incorrect edges, and rendering artifacts.
>
> > W3: Insufficient SFT details
>
> We acknowledge the need for clarity regarding the supervised fine-tuning setup. Our training pipeline follows standard autoregressive generation procedures.
>
> We will make this explicit by adding the following details (already partly present in Appendix B):
> Training uses the HuggingFace SFTTrainer with DeepSpeed ZeRO-3.
>
> * Optimizer: AdamW
> * Learning rate: 5e-4
> * Weight decay: 0.001
> * Batch size: 1 per device, gradient accumulation: 4
> * Warmup ratio: 0.03 with cosine scheduler
> * Number of epochs: 5
> * Loss: categorical cross-entropy over DOT tokens
>
> All code used for SFT is already included in the supplementary material.

---

> > ### Author Response · Authors · 2025-12-02
> >
> > > W4: Missing TikZ results despite related discussion
> >
> > We agree that TikZ-based systems are an important reference point. However, as noted in the paper: DOT emphasizes semantic structure, node–edge relationships, and graph correctness. TikZ emphasizes visual styling and rendering control.
> > For scientific architecture diagrams, workflow systems, computational pipelines, and structured process diagrams, semantic correctness is more important than stylistic layout. DOT is the natural representation for such tasks.
> >
> > TikZ-based works (such as TikZero and Automatikz) target tasks like diagram rendering, not text-to-architecture generation. TikZero focuses on image to TikZ conversion, not text to DOT/TikZ. The modality, target representation, and evaluation pipeline are therefore fundamentally different compared to our problem setting. TikZero is highly relevant to diagram reconstruction, but not directly comparable to our natural language generation setup.
> >
> > Comparison with Automatikz is shown in the table below. It shows that the proposed Text2Arch model performs much better than Automatikz on both the manual annotation set as well as the test set. We have included this now in Appendix R.1 in the revised submission.
> >
> > **Comparison on Text2Arch manual annotation set**
> > |Desc| **ROUGE-L** | **Edit Distance** | **chrF** | **Node Prec** | **Node Recall** | **Node F1** | **Node PR-AUC** | **Edge Prec** | **Edge Recall** | **Edge F1** | **Edge PR-AUC** | **Jaccard Sim.** |
> > |---------------|-------------|--------------------|----------|---------------|-----------------|-------------|------------------|---------------|-----------------|-------------|----------------|---------------|
> > |Automatikz |41.5|531|41.2|46.9|36.9|38.7|12.3|19.1|8.6|10.9|8.9|7.2|
> > |Text2Arch (finetuned DeepSeek-7B)|55.2|407|66.6|66.1|78.1|69.4|27.4|59.4|44.6|49.1|35.1|39.8|
> >
> > **Comparison on Text2Arch test set**
> > |Desc| **ROUGE-L** | **Edit Distance** | **chrF** | **Node Prec** | **Node Recall** | **Node F1** | **Node PR-AUC** | **Edge Prec** | **Edge Recall** | **Edge F1** | **Edge PR-AUC** | **Jaccard Sim.** |
> > |---------------|-------------|--------------------|----------|---------------|-----------------|-------------|------------------|---------------|-----------------|-------------|----------------|---------------|
> > |Automatikz |37.9|608|37.0|42.1|33.6|34.5|11.3|20.4|8.4|10.8|8.3|7.2|
> > |Text2Arch (finetuned DeepSeek-7B)|46.8|608|55.7|66.2|69.6|65.7|21.5|46.4|34.2|38|23.7|28.6|
> >
> > To nevertheless address the reviewer’s suggestion, we conducted a limited transfer study. We took DiagramAgent’s TikZ outputs, used GPT-4 to convert ground-truth DOT to comparable TikZ, and evaluated DiagramAgent. The performance was very poor (even after standardized conversion) confirming that TikZ pipelines struggle with semantic reconstruction:
> >
> > * Test set: Node F1 = 23.1, Edge F1 = 7.2, ROUGE-L = 15.4, CodeBLEU = 11.0
> > * Manual annotation set: Node F1 = 22.1, Edge F1 = 13.3, ROUGE-L = 6.1, CodeBLEU = 12.3
> >
> > These results are significantly below both GPT-4o and our finetuned Text2Arch models.
> >
> > A full TikZ evaluation would require constructing a text-to-TikZ dataset or designing a robust DOT↔TikZ conversion pipeline, which is beyond the scope of the present work, but we appreciate the reviewer’s suggestion and highlight it as an interesting future direction.

---

### Official Review · Reviewer_4mvK · 2025-10-29

**Soundness:** 2
**Presentation:** 3
**Contribution:** 2
**Rating:** 4
**Confidence:** 4

**Summary:**

This paper presents a dataset for diagram generation, based on descriptions. The paper provides a dataset involving the DOT language. It then continues to explore closed-source models such as GPT4o and fine-tuned models, leveraging automatic metrics for evaluation.

**Strengths:**

- a new dataset

- insights on the performance of different models, e.g. fine-tuning helps

- discussion of a relevant new problem, even though there are already some existing works on similar tasks

**Weaknesses:**

- There is no human evaluation at all

- While competitor approaches such as Automatikz are mentioned, there is no comparison

- Some competitor models are missing, e.g. TikZero

- Arguably, just fine-tuning a model on the dataset could be considered a bit incremental in contribution

- I feel more sophisticated automatic metrics such as the ones proposed in TikZero or Automatikz should be explored

TikZero: https://iccv.thecvf.com/virtual/2025/poster/51

**Questions:**

l. 126: I don't understand the argument why you don't want to evaluate the image: comparing the ground-truth image to the generated image is very important from a user perspective

- Why did you exclude GPT5?

- In l.203ff: you do some human filter - where are the agreements? How reliable are the humans involved?

- l.310: so, few-shot prompting is meaingless?

- l.322ff: But SFT mostly leverages training on the training dataset, right?

---

> ### Author Response · Authors · 2025-12-02
> **Response to weaknesses**
>
> We thank the reviewer for their time, providing constructive feedback and for noting several strengths. We address each concern in detail below and incorporate the clarifications and additional results in the revised manuscript.
>
> > W1: Lack of human evaluation
>
> We acknowledge the importance of human evaluation and have conducted a human preference study to assess diagram quality. Across paired comparisons, our model is preferred in 71.6% of the cases while DiagramAgent is preferred in 28.4%. This provides clear evidence that human evaluators perceive the outputs of our system as more semantically correct and structurally coherent. We have included this now in Appendix Q in the revised submission.
>
> > W2: Comparison with Automatikz
>
> The results in the table below show that the proposed Text2Arch model performs much better than Automatikz on both the manual annotation set as well as the test set. We have included this now in Appendix R.1 in the revised submission.
>
> **Comparison on Text2Arch manual annotation set**
> |Desc| **ROUGE-L** | **Edit Distance** | **chrF** | **Node Prec** | **Node Recall** | **Node F1** | **Node PR-AUC** | **Edge Prec** | **Edge Recall** | **Edge F1** | **Edge PR-AUC** | **Jaccard Sim.** |
> |---------------|-------------|--------------------|----------|---------------|-----------------|-------------|------------------|---------------|-----------------|-------------|----------------|---------------|
> |Automatikz |41.5|531|41.2|46.9|36.9|38.7|12.3|19.1|8.6|10.9|8.9|7.2|
> |Text2Arch (finetuned DeepSeek-7B)|55.2|407|66.6|66.1|78.1|69.4|27.4|59.4|44.6|49.1|35.1|39.8|
>
> **Comparison on Text2Arch test set**
> |Desc| **ROUGE-L** | **Edit Distance** | **chrF** | **Node Prec** | **Node Recall** | **Node F1** | **Node PR-AUC** | **Edge Prec** | **Edge Recall** | **Edge F1** | **Edge PR-AUC** | **Jaccard Sim.** |
> |---------------|-------------|--------------------|----------|---------------|-----------------|-------------|------------------|---------------|-----------------|-------------|----------------|---------------|
> |Automatikz |37.9|608|37.0|42.1|33.6|34.5|11.3|20.4|8.4|10.8|8.3|7.2|
> |Text2Arch (finetuned DeepSeek-7B)|46.8|608|55.7|66.2|69.6|65.7|21.5|46.4|34.2|38|23.7|28.6|
>
> > W3: Missing competitor models, including TikZero
>
> TikZero was accepted at ICCV 2025 and focuses on image to TikZ conversion, not text to DOT/TikZ. The modality, target representation, and evaluation pipeline are therefore fundamentally different compared to our problem setting. TikZero is highly relevant to diagram reconstruction, but not directly comparable to our natural language generation setup.
>
> > W4: Concern that fine-tuning may be incremental
>
> Fine-tuning is not the primary contribution of our paper. The central contributions are:
>
> 1. The creation of a large, high-quality, (text, diagram, DOT) dataset for scientific architecture diagrams.
>
> 2. A new benchmark for text-to-structured-diagram generation with graph-based evaluation.
>
> 3. Empirical evidence demonstrating that domain-aligned supervision enables smaller models to outperform significantly larger models.
>
> Fine-tuning is a necessary and standard component of evaluating dataset utility but not the novelty of the work.
>
> > W5: Sophisticated automatic metrics (TikZero, Automatikz)
>
> The evaluation metrics in TikZero and Automatikz are designed for image-based TikZ evaluation, not for graph-structured DOT generation. Our task is purely structural. We therefore rely on graph-level metrics consistent with DiagramAgent, including node and edge precision, recall, F1, Jaccard, and structural equivalence. These metrics are directly aligned with the semantics of DOT code.
> Since we generate DOT and render images only for visualization, image similarity metrics are not meaningful for our setting.

---

> > ### Author Response · Authors · 2025-12-02
> > **Response to Questions**
> >
> > > Q1: Why we do not evaluate image similarity
> >
> > Our output is DOT code, and the scientific relevance lies in whether the code captures the correct structure: node contents, directed edges, and logical relationships. Rendering the DOT to an image determines only layout and visual aesthetics, not correctness of the underlying architecture.
> >
> > Thus, image-level comparison would primarily evaluate layout differences, which are semantically irrelevant. Instead, we evaluate structural equivalence. The focus of the paper is on accurate generation of the underlying graph, not on stylistic rendering.
> >
> > > Q2: Why GPT-5 was excluded
> >
> > GPT-5 was released just before the ICLR submission deadline. GPT-5.1 was also released post-deadline. For fairness and compliance, we included models readily available before the submission deadline. We have included this now in Appendix S.3 in the revised submission.
> >
> > > Q3: Human filtering and annotation reliability
> >
> > Training data was filtered by annotators with deep domain knowledge (the authors), and the task is highly objective. For the manually annotated evaluation subset, the annotators labeled node and edge sets with very high consistency: 98 percent agreement for nodes and 96 percent for edges. Due to the cost of expert annotation, we did not perform overlapping annotation for the entire dataset, but the agreement numbers above reflect strong reliability. We have included this now in Appendix Q in the revised submission.
> >
> > > Q4: Are few-shot results meaningless?
> >
> > Few-shot prompting was ineffective for this task. With base and instruction-tuned models, few-shot prompting often led to one of two failure modes:
> > * repetition of the few-shot examples,
> > * empty or syntactically invalid DOT outputs.
> >
> > This behavior is consistent with the gap between LLM pretraining (optimized for natural-language continuation) and the grammar-constrained nature of DOT code generation. Fine-tuning aligns the model to the structured output domain and yields consistent improvements, as shown in Tables 1 and 2.
> >
> > > Q5: SFT and reliance on the training dataset
> >
> > Yes, our supervised fine-tuning uses the provided training set combined with a consistent system prompt. The loss is computed on output tokens following standard next-token prediction. Fine-tuning adapts the model to the structural and syntactic properties of DOT code, which cannot be learned reliably from few-shot examples alone.

---

### Official Review · Reviewer_pUFM · 2025-11-01

**Soundness:** 2
**Presentation:** 1
**Contribution:** 2
**Rating:** 0
**Confidence:** 4

**Summary:**

The paper introduces a dataset for training models to automatically convert natural-language descriptions into scientific architecture diagrams. The authors report that a compact model trained on the proposed data outperforms larger models (e.g., GPT-4o) on the benchmark tasks defined in the paper.

**Strengths:**

1. Framing NL to diagram generation for scientific architectures is a well-scoped and timely problem with practical value for documentation and education.
2. The paper provides a purpose-built dataset aligned with the task, which can catalyze further research and standardized evaluation.
3. Initial experiments indicate that a smaller, task-specialized model can surpass much larger general-purpose models, suggesting meaningful gains from domain-specific supervision.

**Weaknesses:**

1. The paper’s core innovations and their separation from prior art are not sufficiently explicit.
2. The definition of scientific architecture, the selection of the 99 images in Fig. 3, and the complexity distribution of diagrams across the full dataset are unclear.
3. It is unclear whether a diagram has a single “correct” \texttt{DOT} representation, and how correctness is measured when multiple valid encodings exist.
4. For the same digram, only one correct DOT exist? If not, how they meature the correctness?
5. If many diagrams resemble the simple patterns in Fig. 1, current VLMs may already perform well.

**Questions:**

1. Definition and coverage of “scientific architecture.”
2. How were the 99 images in Fig. 3 selected, and how does their complexity compare with the full dataset?
3. Is the target \texttt{DOT} constrained by a formal grammar during training/inference?
4. How robust is the model to paraphrase, long/underspecified descriptions, or OOD domains?

---

> ### Author Response · Authors · 2025-12-02
> **Response to weaknesses**
>
> We thank the reviewer for the detailed assessment and for highlighting both strengths and areas for improvement. We also incorporate the clarifications and additional results in the revised manuscript.
>
> We respectfully believe that a **rating of 0** does not fully reflect the contributions, novelty, and empirical depth of the submission, especially given the reviewer’s own recognition of the task’s relevance, dataset value, and the competitive performance of our compact model. We address the various points below.
>
> > W1: Core innovations and separation from prior work
>
> The submission contributes two main advances that, to our knowledge, are not present in prior work:
> We introduce TEXT2ARCH, a large-scale dataset of more than 75K examples containing architecture diagrams, clean natural language descriptions, and their corresponding DOT graphs. Existing diagram-related datasets do not provide consistently aligned (text, diagram, code) triplets, nor do they include domain-specific descriptions that accurately reflect scientific architectural content. TEXT2ARCH fills this gap by providing high-quality, standardized supervision at scale.
>
>
> We formulate and study a new task: generating scientific architecture diagrams from natural language descriptions through intermediate DOT code. Prior work focuses on diagram recognition, VLM-based image understanding, or free-form graphics generation. The text-to-DOT formulation is new, and we show that domain-specific supervision enables small models to outperform substantially larger general-purpose models.
>
> > W2: Definition of “scientific architecture” and dataset complexity
>
> We use the term “scientific architecture diagrams” to refer to structured graphical representations of computational or experimental workflows, methodological pipelines, and scientific system designs commonly found in scientific and engineering publications. We have included this now in Appendix S.1 in the revised submission.
>
> Figure 3 provides the complexity distribution of diagrams across the dataset in terms of nodes and edges. The train, validation, and test splits exhibit closely aligned distributions with an average of 15.24 nodes and 13.89 edges per graph.
>
> As noted in the submission, 54.3 percent of samples are labeled as easy, 32.4 percent as medium, and the remainder as hard. The manually annotated set of 99 diagrams contains 50 easy, 30 medium, and 19 hard examples, which closely matches the full dataset distribution. This ensures that examples selected for qualitative analysis are representative of the overall dataset.
>
> > W3: Multiple valid DOT representations and correctness measurement
>
> We appreciate the reviewer’s observation. A single diagram can indeed correspond to multiple syntactically different DOT files due to variations in node ordering, edge ordering, and layout directives. To provide a unique training target and ensure valid comparisons, we canonicalize all DOT graphs prior to use.
> Specifically, we:
>
> (i) sort nodes lexicographically by label,
>
> (ii) sort edges by source–target index pairs,
>
> (iii) remove layout or stylistic attributes that do not affect topology.
>
> During evaluation, we compute structural equivalence by converting DOT graphs into adjacency matrices and performing graph isomorphism checks. Two diagrams are considered equivalent if their node labels and directed edges correspond under a bijection. All reported graph-level metrics operate strictly on node content and edge connectivity and are invariant to layout differences.
>
> This approach is consistent with evaluation strategies used in other structured-generation domains and additionally, our text based evaluations align with the evaluations adopted by DiagramAgent (CVPR 2025), which we use as a baseline. We have included this now in Appendix S.2 in the revised submission.
>
> > W4: Whether a diagram has one “correct” DOT
>
> Following the canonicalization procedure, each diagram is mapped to a single DOT representation that uniquely captures its structure. This ensures consistent supervision and unambiguous evaluation, while remaining correct up to graph isomorphism. The evaluation metrics are therefore insensitive to semantically irrelevant syntactic variations. We have included this now in Appendix S.2 in the revised submission.
>
> > W5: On whether VLMs may already perform well on simple diagrams
>
> We want to clarify a misunderstanding: this is not a vision-language modeling problem. The image shown in Fig. 1a is included solely for illustrative purposes. The input to our system is the textual description (Fig. 1b), not the image.
> The task is natural language to architecture diagram generation, where the output is DOT code. VLMs are not suitable for this setting because they require an image input and do not perform text-to-text structured generation. Diffusion models can generate images, but they struggle with textual fidelity, logical structuring, and editability. These limitations motivate our text-to-DOT formulation.

---

> ### Author Response · Authors · 2025-12-02
> **Response to Questions**
>
> > Q1 and Q2
>
> Same as W1 and W2. Already answered above.
>
> > Q3: DOT grammar and constraints during training and inference
>
> All ground truth DOT files follow the Graphviz DOT specification. We do not employ constrained decoding at inference because DOT validity can only be checked once the sequence is complete. However, we adopt a context-free grammar aligned with the DOT specification for validation during training-time preprocessing. All manually annotated DOT outputs also follow this specification.
>
> > Q4: Robustness to paraphrasing, longer descriptions, and OOD linguistic variations
>
> To assess robustness, we generated lengthened and shortened paraphrased variants of the textual descriptions using GPT4o-1120 (prompt included in Appendix P). The results are summarized in the table below for our best model (finetuned DeepSeek-7B). We also include results in Tables 5 and 6 in Appendix P in the revised paper. As expected, significant length shifts degrade performance, but the trends are consistent and provide valuable insights into the sensitivity of text-to-structure generation. In the manual annotations set, the original descriptions contain ~201 words, while lengthened and shortened paraphrased variants contain 599 and 139 words respectively.
>
> **Description length variation results on Text2Arch manual annotation set**
>
> |Desc| **ROUGE-L** | **CodeBLEU** | **Edit Distance** | **chrF** | **Node Prec** | **Node Recall** | **Node F1** | **Node PR-AUC** | **Edge Prec** | **Edge Recall** | **Edge F1** | **Edge PR-AUC** | **Jaccard Sim.** |
> |---------------|-------------|-------------|--------------------|----------|---------------|-----------------|-------------|------------------|---------------|-----------------|-------------|----------------|---------------|
> |Lengthened Desc|44.7|37.2|682|48.7|72.8|62.7|64.4|23.8|52.3|35.0|40.2|25.1|31.2|
> |Shortened Desc|55.3|50.0|491|65.0|70.6|75.5|71.4|19.8|61.9|45.3|50.7|34.0|39.6|
> |Original Desc.|55.2|49.3|407|66.6|66.1|78.1|69.4|27.4|59.4|44.6|49.1|35.1|39.8|
>
> We appreciate the reviewer’s time and feedback. We believe the paper introduces a meaningful new dataset and task, provides clear empirical evidence of the benefits of domain-specific supervision, and offers reproducible methodology and evaluation. We hope that the clarifications and additional analyses above help reassess the technical contribution and justify a rating higher than 0.

---

### Meta-Review · Area_Chair_vJex · 2026-01-07

**Summary:**

Reviewers agree that TEXT2ARCH addresses a well-scoped and practically valuable problem by introducing a large, purpose-built dataset for converting natural-language descriptions into scientific architecture diagrams via DOT code. The main concerns driving mixed recommendations were clarity and positioning of the core contributions, evaluation rigor (human evaluation, baseline fairness, robustness, and reproducibility), and potential circularity or bias introduced by GPT-assisted curation and evaluation. The rebuttal is extensive and materially strengthens the paper by clarifying task definitions, dataset complexity, and correctness criteria; adding human evaluation, compilation-rate analysis, stronger baselines (Automatikz), robustness studies, and detailed SFT specifications. Remaining disagreements are largely about philosophical scope (DOT vs TikZ, dataset vs method novelty) and the feasibility of even larger human-labeled or cross-representation evaluations, rather than unresolved technical flaws.

**Reviewer Concerns:**

Reviewer pUFM: The rebuttal addresses all substantive concerns by clearly defining “scientific architecture,” justifying dataset complexity and sample selection, explaining canonicalization and graph-isomorphism-based evaluation, clarifying that the task is text-to-DOT rather than VLM-based, and adding robustness experiments on paraphrasing and length variation.

Reviewer 4mvK: The rebuttal directly addresses the lack of human evaluation, missing Automatikz comparisons, concerns about incremental fine-tuning, metric choice, annotation reliability, and model exclusions by adding human preference studies, Automatikz results, annotation agreement statistics, and clearer methodological explanations.

Reviewer uvQ1: The rebuttal resolves most concerns by adding DOT compilation success rates, richer qualitative visual comparisons, detailed SFT hyperparameters, and a principled justification (with auxiliary experiments) for excluding TikZ-native evaluation, leaving only minor scope limitations.

Reviewer yJFJ: The rebuttal substantially addresses baseline fairness, evaluation circularity, matching thresholds, and robustness by adding TikZ-domain checks, GPT-independence clarifications, multi-threshold analyses, and expanded appendices, while acknowledging but reasonably justifying limits on human annotation scale and full DOT1/DOT2 retraining.

**Reviewer Scores:**

Reviewer pUFM (0) did not state a score change, but since all of their concrete questions and criticisms were directly and thoroughly addressed, their concerns appear fully resolved by the rebuttal.

Reviewer 4mvK (4) did not explicitly indicate a score update, and given that all major weaknesses were addressed with new experiments and analyses, the rebuttal largely resolves their concerns.

Reviewer uvQ1 (6) explicitly stated openness to revising their score, and since each listed weakness was answered with new empirical evidence or clarification, their concerns appear largely resolved.

Reviewer yJFJ (6) did not state a score change, but because the rebuttal responds in detail to nearly all technical and methodological questions, their concerns are mostly resolved, with only acknowledged scope and resource limitations remaining.

---

### Decision · Program_Chairs · 2026-01-26

Accept (Poster)